# Duplex DNA engagement and RPA oppositely regulate the DNA-unwinding rate of CMG helicase

Hazal B. Kose[1], Sherry Xie[1], George Cameron[1], Melania S. Strycharska[1] & Hasan Yardimci [1✉]

A ring-shaped helicase unwinds DNA during chromosome replication in all organisms. Replicative helicases generally unwind duplex DNA an order of magnitude slower compared to their in vivo replication fork rates. However, the origin of slow DNA unwinding rates by replicative helicases and the mechanism by which other replication components increase helicase speed are unclear. Here, we demonstrate that engagement of the eukaryotic CMG helicase with template DNA at the replication fork impairs its helicase activity, which is alleviated by binding of the single-stranded DNA binding protein, RPA, to the excluded DNA strand. Intriguingly, we found that, when stalled due to interaction with the parental duplex, DNA rezipping-induced helicase backtracking reestablishes productive helicase-fork engagement, underscoring the significance of plasticity in helicase action. Our work provides a mechanistic basis for relatively slow duplex unwinding by replicative helicases and explains how replisome components that interact with the excluded DNA strand stimulate fork rates.

---

[1] Single Molecule Imaging of Genome Duplication and Maintenance Laboratory, The Francis Crick Institute, NW1 1AT London, UK.
✉email: Hasan.Yardimci@crick.ac.uk

All cells utilize a ring-shaped helicase that separates the two strands of the DNA double helix during chromosome replication. Replicative helicases form a homohexameric complex such as gp4 in bacteriophage T7, DnaB in bacteria, large T antigen in Simian Virus 40 (SV40), E1 helicase in bovine papillomavirus, and minichromosome maintenance (MCM) in archaea[1]. The only known exception to the homohexameric nature is the eukaryotic replicative DNA helicase, comprising the Mcm2-7 motor containing six different but highly related AAA + ATPases. In G1 phase, Mcm2-7 rings are loaded onto duplex DNA as double hexamers[2–5]. Activation of the helicase in S phase occurs upon binding of Cdc45 and GINS, and subsequent remodeling of Mcm2-7 from encircling double-stranded (ds) to single-stranded (ss) DNA in its central channel[6–8]. Through ATP hydrolysis, Cdc45/Mcm2-7/GINS (CMG) complex translocates along the leading-strand template in the 3′–5′ direction and unwinds DNA at the replication fork[6]. In addition to its role in DNA unwinding, the replicative helicase acts as a hub to organize other replication factors around itself, thus assembling the replisome.

CMG was first characterized biochemically in isolation by purifying the complex from *Drosophila* embryo extracts[9]. Recombinant *Drosophila*, yeast and human CMG complexes were later shown to unwind DNA substrates containing Y-shaped fork structures (fork DNA) in an ATP-dependent manner[10–12]. When unwinding DNA at the fork, isolated CMG can freely bypass protein obstacles on the lagging-strand template, indicating that this strand is excluded from the helicase central channel during translocation[7]. Thus, CMG functions via steric exclusion, a mechanism shared by all known replicative helicases[6,13–16].

Using single-molecule magnetic tweezers, we earlier found that individual *Drosophila* CMG complexes exhibit forward and backward motion while unwinding dsDNA[17], similar to E1 and T7 gp4 helicases[15,18,19]. Furthermore, the helicase often enters long-lived paused states, leading to an average unwinding rate of 0.1–0.5 base pairs per second (bp s$^{-1}$), which is approximately two orders of magnitude slower than eukaryotic replication fork rates observed in vivo[20,21]. However, recent single-molecule work with yeast CMG suggests that the helicase translocates on ssDNA at 5–10 bp s$^{-1}$ [22]. Single-molecule trajectories by other replicative helicases such as DnaB and gp4 suggest that helicase pausing during dsDNA unwinding is a general property of these enzymes[18,23]. However, it is not clear why replicative helicases frequently halt whilst moving at the fork and how higher speeds are achieved by the entire replisome.

The rate of DNA unwinding by *Escherichia coli* DnaB and T7 gp4 is substantially enhanced when engaged with their corresponding replicative polymerases[24,25], suggesting that the rate of fork progression in eukaryotes may also depend on DNA synthesis. Likewise, uncoupling of CMG from the leading-strand polymerase leads to fork slowing in an in vitro purified yeast system[26]. Accordingly, 5–10-fold reduction in helicase speed was observed in *Xenopus* egg extracts when DNA synthesis was inhibited[27]. However, this decrease in CMG translocation rate is not sufficient to account for the ~100-fold lower rates seen in DNA unwinding by isolated CMG[17]. Thus, in addition to polymerases, other replisome-associated factors may be essential to increase the rate of DNA unwinding by the helicase. Intriguingly, single-molecule visualization of the ssDNA-binding protein RPA during CMG-driven DNA unwinding indicated that *Drosophila* CMG proceeds at an average rate of 8 bp s$^{-1}$ at the fork[7]. This result suggests that binding of RPA to unwound DNA improves the rate of translocation by CMG. One possible explanation for RPA-induced rate increase is the association of RPA with the translocation strand behind CMG and concomitant hindrance of helicase backtracking. In addition, RPA binding to the excluded strand may prevent DNA reannealing in front of the helicase, thus increasing the rate of unwinding. Finally, RPA binding may also influence helicase activity by altering the interaction of CMG with the excluded strand.

Control of DNA unwinding by replicative helicases through their interaction with the excluded strand has been demonstrated in different organisms. Although wrapping of the displaced strand around an archaeal MCM was proposed to increase its helicase activity[28], interaction of DnaB with the displaced strand through its exterior surface adversely affects DNA unwinding[29]. While it is not clear whether CMG makes contacts with the lagging-strand template via specific residues on its outer surface, the presence of the excluded strand is important for unwinding of dsDNA by CMG. Notably, unwinding of synthetic DNA substrates by CMG relies not only on the availability of a 3′ ssDNA tail for CMG binding, but also on the presence of a 5′ overhang. On partially duplexed DNA lacking the 5′ flap, CMG binds the 3′ ssDNA and subsequently slides onto dsDNA upon meeting the duplexed region[11,30]. Equally, T7 gp4, DnaB, and archaeal MCM transfer from translocating on ssDNA to dsDNA without unwinding the template when encountering a flush ss-dsDNA junction[14,31–33]. This unproductive translocation on duplex DNA is likely a consequence of the central pores of these motors being sufficiently large to accommodate dsDNA.

In this study, we sought to address how the interaction of CMG with DNA at the fork regulates its helicase activity and the mechanism by which RPA stimulates DNA-unwinding rate of the eukaryotic replicative helicase.

## Results

**Direct visualization of RPA-facilitated unwinding by CMG.** We previously demonstrated CMG-driven unwinding of surface-immobilized 2.7-kb dsDNA substrates through accumulation of EGFP-tagged RPA (EGFP-RPA) using total internal reflection fluorescence (TIRF) microscopy[7]. To more directly assess the processivity and rate of DNA unwinding by CMG, we examined the translocation of fluorescently-labeled CMG complex on 10-kb linear dsDNA molecules containing a forked end (Fig. 1a). DNA molecules were tethered to the glass surface at the forked end and to a microsphere at the opposite end. The 3′ tail of the fork contained a 40-nt polyT ssDNA (dT$_{40}$) for CMG binding and a Cy3 fluorophore to follow the position of the translocation strand (Fig. 1a). After immobilizing DNA on the surface (Fig. 1b), fluorescent CMG (CMG$^{LD655}$) was drawn into the flow cell and incubated in the presence of ATPγS. Subsequently, the CMG-ATPγS mixture in the flow channel was exchanged with a solution containing ATP and EGFP-RPA (Fig. 1c). Near-TIRF imaging was performed in the absence of buffer flow through the flow cell. We observed EGFP-RPA binding initially at the forked end of the stretched DNA before growing as a linear tract towards the microsphere-tethered end (Fig. 1d, left panel). The leading-strand template bound by EGFP-RPA compacted and appeared as a diffraction-limited spot moving at the fork. When either CMG from ATPγS-containing buffer or ATP from subsequent RPA-supplemented solution were omitted, we did not detect linear EGFP-RPA tracts indicating that we are visualizing CMG-dependent fork unwinding. Approximately 25% of RPA tracts contained labeled CMG translocating at the fork (Fig. 1d, middle panel). The relatively low fraction of labeled CMG molecules is most likely owing to inefficient conjugation of the fluorophore or subsequent photobleaching. The Cy3-labeled translocation strand colocalized with the bright EGFP-RPA spot as expected (Fig. 1d, right panel). Many 10-kb DNA molecules were unwound entirely at an average rate of 4.5 ± 1.6 bp s$^{-1}$ (mean ± SD) by CMG (Fig. 1e). The majority of molecules were not fully unwound,

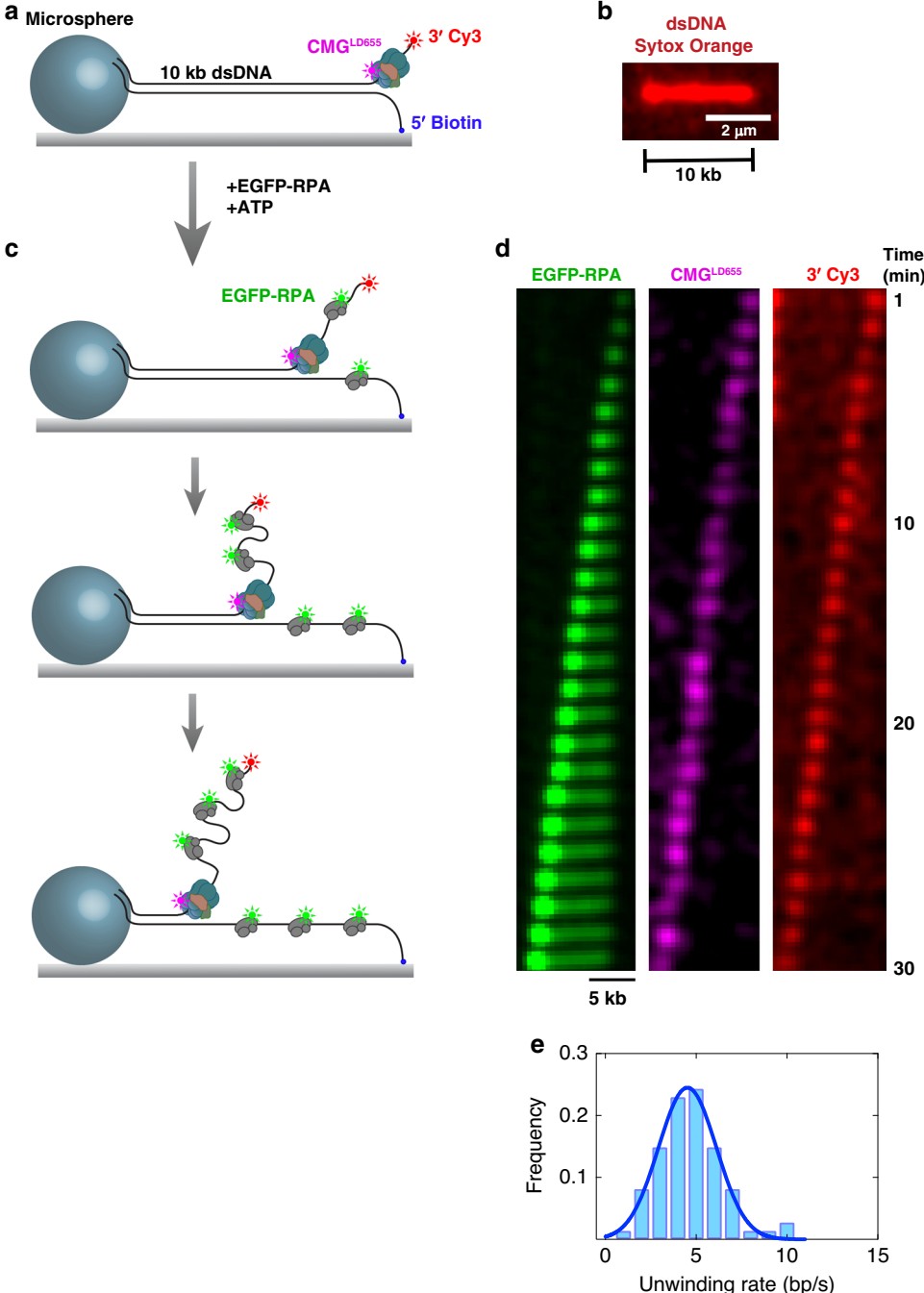

**Fig. 1 Direct visualization of RPA-facilitated processive fork unwinding by individual CMG molecules. a** A 10-kb fragment of λ DNA was ligated to a short fork DNA substrate at one end and to a digoxigenin-modified DNA fragment on the opposite end. The 5' tail of the forked end contained a biotin to attach the 10-kb substrate to biotin-functionalized glass through biotin-streptavidin binding. The digoxigenin-modified end was coupled to anti-digoxigenin-coated microsphere to stretch DNA molecules by buffer flow, and to subsequently attach this end to the surface (details are described in the Methods section). The DNA substrate was labeled with Cy3 at the 3' dT$_{40}$ ssDNA tail near the fork junction. LD655-labeled CMG was bound to the dT$_{40}$ ssDNA in the presence of ATPγS. **b** A sample stretched 10-kb linear DNA stained with a fluorescent dsDNA intercalator Sytox Orange. Average length of stretched DNA molecules were determined by measuring the end-to-end distance of 717 individual DNA molecules ($n = 2$ independent experiments). **c** After binding CMG on the surface-immobilized DNA, EGFP-RPA, and ATP was introduced to initiate unwinding. While CMG unwinds DNA at the fork, EGFP-RPA binds both strands of unwound DNA. **d** Kymograph showing a representative 10-kb DNA being unwound entirely by a single CMG complex. EGFP-RPA (left panel), LD655-labeled CMG (center panel), and 3' Cy3 (right panel) are imaged during unwinding under near-TIRF conditions. Images were acquired in the absence of buffer flow. **e** Histogram of CMG-catalyzed DNA-unwinding rates on stretched DNA (74 individual DNA molecules analyzed from two independent experiments). Source data are provided as a Source Data file.

likely being interrupted because of CMG encountering nicks present on stretched DNA. On some molecules, the unwound region of the RPA-coated leading-strand template either dissociated from (Supplementary Fig. 1a) or diffused along (Supplementary Fig. 1b) the stretched lagging-strand template. We attribute these observations to CMG encountering a nick on the leading-strand template. Labeled CMG always dissociated from DNA upon hitting a leading-strand nick suggesting that the helicase ran off the free 5′-end of the translocation strand (Supplementary Fig. 1a, b). In other cases, partially unwound stretched DNA broke, suggesting that CMG ran into a nick on the lagging-strand template (Supplementary Fig. 1c). Although the unwound lagging-strand template instantly moved to the surface-tethered point, the unwound leading-strand template and CMG moved towards the microsphere (Supplementary Fig. 1c). In our previous study[7], we used non-stretched DNA molecules and measured unwinding rates relying solely on the level of EGFP-signal intensity. Therefore, any event where CMG encountered a nick was scored as the entire DNA being unwound, which may have led to an overestimation of unwinding rate (8.2 bp s$^{-1}$ as opposed to 4.5 bp s$^{-1}$). Together, our results clearly demonstrate that when RPA is available, individual CMG helicases can unwind thousands of base pairs of dsDNA at a rate matching the ssDNA translocation rate of the helicase[22].

**RPA increases the rate of DNA unwinding by CMG**. The rate of duplex unwinding by CMG alone[17] was found to be strikingly low (0.1–0.5 bp s$^{-1}$) compared with that measured by single-molecule fluorescence imaging in the presence of RPA (Fig. 1). Therefore, the translocation speed of CMG that unwinds dsDNA at the fork must be stimulated by an order of magnitude by RPA. Because these measurements were done using two different experimental methods under different conditions, we compared the rate of CMG translocation at the fork with and without RPA using a single assay. We examined CMG helicase activity at the ensemble level on a fork DNA substrate containing 10 repeats of GGCA sequence, d(GGCA)$_{10}$, on the 5′ lagging-strand arm, dT$_{40}$ as the 3′ arm, and 236 bp of dsDNA (Fig. 2a). CMG was first incubated with the fork substrate in the presence of ATPγS for its binding to the 3′ tail. The d(GGCA)$_{10}$ sequence folds into secondary hairpin-like structures and prevents CMG binding to this strand[34]. We confirmed that RPA alone did not unwind this DNA substrate in the absence of CMG (Fig. 2a, lane 2). After CMG binding, ATP was added to the reaction to trigger helicase translocation. When RPA was added simultaneously with ATP, we observed significant unwinding of the substrate in a CMG-dependent manner (Fig. 2a, lane 4). In contrast, no unwinding was detected when RPA was omitted from the reaction (Fig. 2a, lane 3), most likely owing to reannealing of the complementary strands behind the helicase. To overcome this limitation, we generated a similar fork substrate that contained a Cy5-labeled oligonucleotide downstream of a 252 bp duplex. The Cy5-labeled strand contained a d(GGCA)$_{10}$ 5′ tail and 28-nt complementary sequence to the translocation strand (Fig. 2b). Therefore, CMG should displace the Cy5-modified strand if it can translocate through the 252-bp dsDNA even if DNA rewinds in the wake of the advancing helicase. To measure the kinetics of CMG translocation, the reaction was quenched at different times after ATP addition. When RPA was added together with ATP, CMG rapidly displaced the Cy5-labeled strand (Fig. 2c). To quantify CMG-dependent unwinding, the data were corrected for Cy5-modified strand displaced by RPA alone (Supplementary Fig. 2a). CMG displaced the Cy5-modified strand to a significant extent even in the absence of RPA (Fig. 2d), indicating that lack of strand separation on the 236-bp duplex fork (Fig. 2a) was due to

rewinding of DNA trailing the helicase. Unwinding of the 28-bp duplex region was dependent on CMG binding to the upstream 3′ dT$_{40}$ ssDNA tail (Supplementary Fig. 2b), indicating that the helicase translocated through the 252-bp duplex before displacing the Cy5-modified strand. In the presence of RPA, CMG unwound all 280 bp (252 bp + 28 bp) at an observed rate of $k_{obs} = 1.31 \pm 0.11$ min$^{-1}$ (Fig. 2e, solid line), leading to a DNA-unwinding rate of $6.1 \pm 0.5$ bp s$^{-1}$ (280 bp × $k_{obs}$), in good agreement with single-molecule measurements (Fig. 1). In contrast, CMG alone displaced the Cy5-modified strand at a rate of $k_{obs} = 0.048 \pm 0.003$ min$^{-1}$, indicating a DNA-unwinding rate of $0.22 \pm 0.01$ bp s$^{-1}$ (Fig. 2f, solid line), consistent with linear unwinding rates measured with magnetic tweezers[17]. Therefore, the gel-based helicase assays described here suggest that RPA increases the rate of CMG translocation at the fork about an order of magnitude, in agreement with single-molecule studies. Importantly, because DNA reannealed behind the helicase when unwinding long duplex DNA in the absence of RPA (Fig. 2a, c), the stimulation in helicase speed by RPA is unlikely due to the prevention of helicase backtracking.

**Duplex engagement at the fork impedes CMG helicase activity**. Our magnetic tweezers measurements indicated that the slow nature of DNA unwinding by isolated CMG was owing to frequent entry of the helicase into long-lived paused states[17]. We wished to determine the origin of these pausing events. Given the ability of CMG to encircle dsDNA, we considered the possibility that while translocating along the leading-strand template, the helicase may engage with the parental duplex at the fork junction as an off-pathway interaction. Inhibition of helicase activity was reported for T7 gp4 owing to its interaction with the parental dsDNA on fork DNA substrates[33]. In addition, enhanced DnaB helicase activity upon duplexing the excluded strand of fork DNA has been attributed to a decrease in probability of the helicase encircling parental duplex DNA at the fork[14,35]. In support of this model, constricting the central channel of DnaB by point mutations led to elevated levels of fork DNA unwinding[36]. Likewise, while unwinding dsDNA, CMG may frequently slip onto the fork nexus to capture a fragment of duplex DNA in its central channel and potentially enter into a non-translocating state. Based on this model, duplexing and thus stiffening and enlarging the displaced arm of fork DNA should stimulate DNA unwinding by impeding the helicase ring encircling duplex DNA at the fork junction, as seen for DnaB[14]. In previous work, we used fork substrates with d(GGCA)$_{10}$ on the 5′ lagging-strand arm[7]. As this sequence folds into secondary structures, it may prevent CMG from partially encircling the parental duplex at the fork junction. We found that replacing d(GGCA)$_{10}$ on the 5′ tail with dT$_{40}$ led to twofold decrease in DNA unwinding by CMG (Supplementary Fig. 3a) in line with the prediction that duplex engagement at the fork junction impairs CMG translocation. We tested this model in more detail using a fluorescence-based single-turnover kinetic unwinding assay[7]. The fork DNA templates used in this assay contained 28-bp duplex DNA, a dT$_{40}$ leading-strand arm and 22-nt long either single-stranded (Fork$^{ssLag}$) or duplexed lagging-strand arm (Fork$^{dsLag}$) (Fig. 3a and Supplementary Fig. 3b). Fork substrates were modified with Cy5 at the 5′-end of the leading-strand template and a BHQ2 quencher on the complementary strand so that strand separation leads to fluorescence increase (Fig. 3a). To detect single-turnover unwinding kinetics, CMG was pre-bound to the fork in the presence of ATPγS, and subsequently ATP was added together with a competitor oligonucleotide to sequester free CMG (Supplementary Fig. 3c). We confirmed that the fluorescence increase was dependent on CMG pre-binding as well as subsequent addition of ATP on the 28-bp

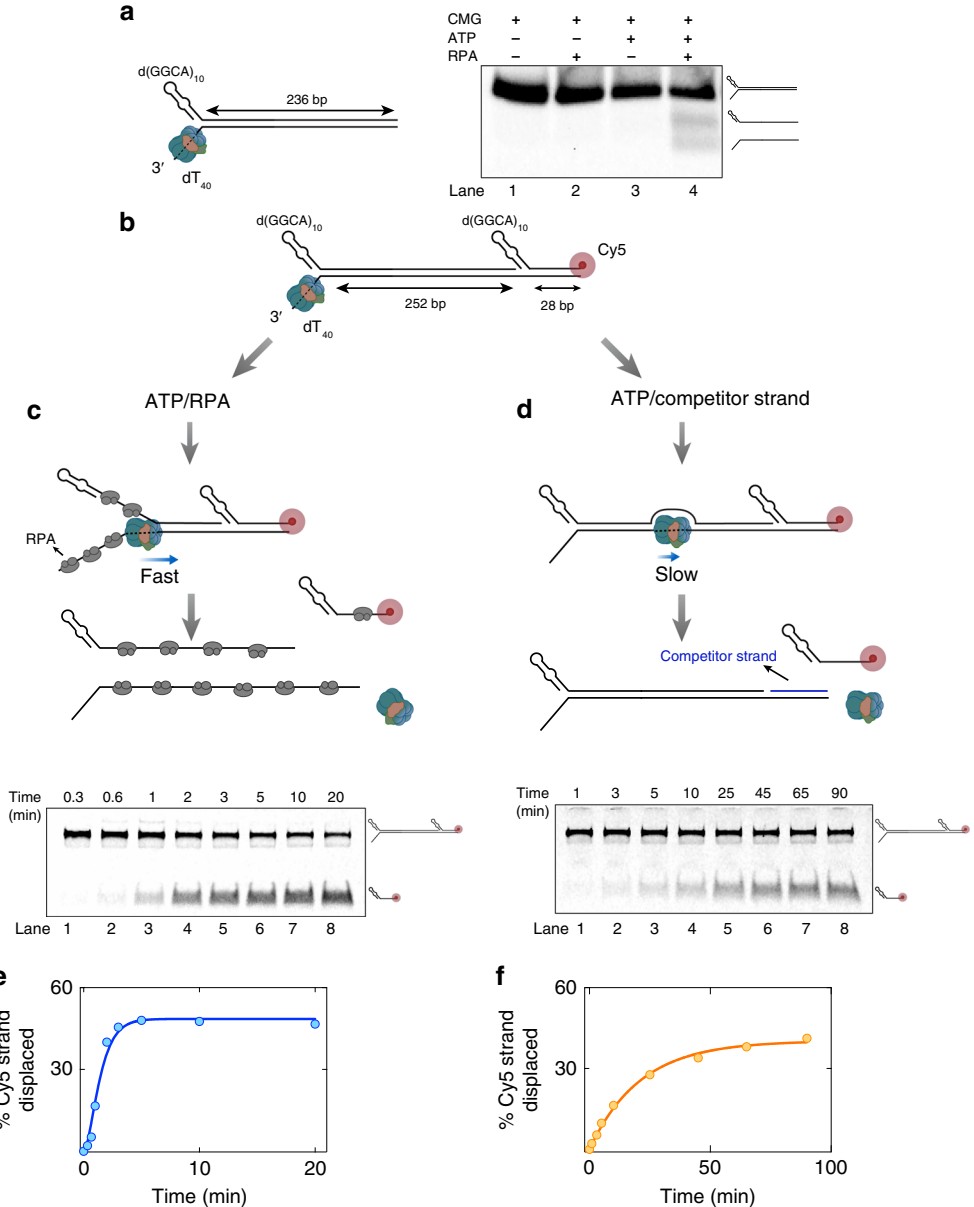

**Fig. 2 RPA increases the rate of DNA unwinding by CMG. a** Fork DNA containing 236-bp dsDNA was incubated with CMG in the presence of ATPγS for binding to 3′ $dT_{40}$ tail. ATP was added with (lane 4) or without (lane 3) RPA and incubated further before separating on 3% agarose. DNA was labeled internally with multiple Cy5 fluorophores on both strands. This experiment was performed once. **b** Fork DNA substrate containing 252-bp long dsDNA, followed by 28-bp duplex and Cy5-modification on the excluded strand was bound by CMG. ATP was added to initiate translocation by the helicase. **c** CMG-mediated displacement of Cy5-labeled strand in the presence of RPA. When included in ATP buffer, RPA prevents reannealing of DNA behind the helicase as well as new CMG binding. **d** CMG-mediated displacement of Cy5-labeled strand in the absence of RPA. DNA rewinds within the 252-bp duplex region. To prevent rehybridization of the Cy5-labeled strand to long DNA substrate, excess competitor oligonucleotide containing complementary 28-nt sequence to long DNA was added with ATP. To achieve single-turnover kinetics, excess $dT_{40}$ oligonucleotide was included in ATP buffer that captures any free CMG. **e, f** Percentage of Cy5-modified strand displaced versus time by CMG in the presence **e** and absence of RPA **f**. The data in **c–f** represent mean from $n = 2$ independent experiments, and were fit to Eq. (1) with $m = 1$ or 2 (see Methods) resulting in best $R^2$ value. Solid lines are fits to Eqs. (2) and (3) in **e** and **f**, respectively. Source data are provided as a Source Data file.

duplex fork (Supplementary Fig. 3d). Importantly, Fork$^{dsLag}$ showed higher fluorescence increase than Fork$^{ssLag}$ (Fig. 3b), indicating a greater level of unwinding consistent with our hypothesis. To ensure this effect is not owing to a discrepancy in the efficiency of helicase binding the two substrates, CMG was first bound to Fork$^{ssLag}$ before triggering unwinding with ATP in the presence of an additional oligonucleotide complementary to the lagging-strand tail (Comp$^{LagTail}$). The addition of Comp$^{LagTail}$ with ATP was sufficient to increase the level of fork

unwinding (Supplementary Fig. 3e), demonstrating that duplexing the 5′ tail stimulates CMG helicase activity rather than proper fork binding. However, we cannot exclude the possibility that CMG encircles both the 3′- and 5′-ssDNA tails in its central channel when bound to the fork substrate with ATPγS and that adding Comp$^{LagTail}$ may configure CMG to exclude the 5′ tail from the helicase pore.

To rule out the possibility that improved unwinding of Fork$^{dsLag}$ is an inherent feature of the DNA template and not

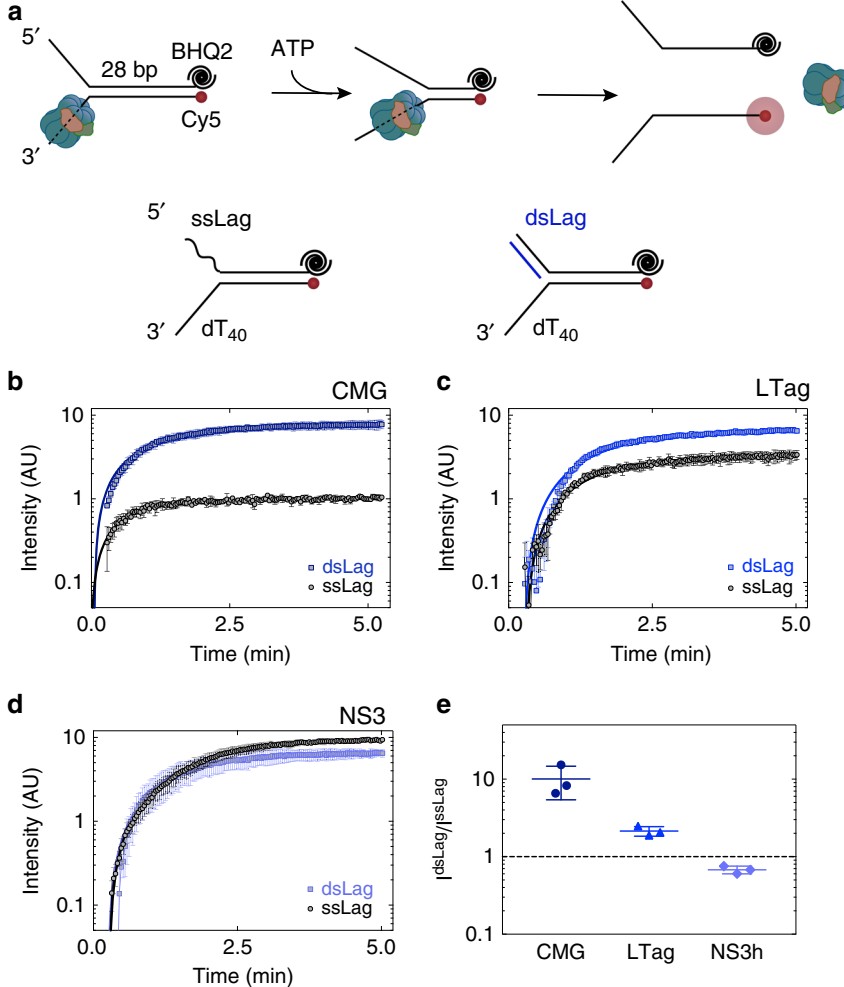

**Fig. 3 Duplex DNA engagement by CMG at the fork impedes its helicase activity. a–d** Single-turnover unwinding of fork DNA substrates containing ss (black) or ds (blue) lagging-strand arm by CMG **b**, large T antigen **c**, and NS3 **d**. DNA substrates contained 28-bp duplex region and were modified with Black Hole Quencher 2 (BHQ2) at the 3′-end of the lagging-strand template and Cy5 at the 5′-end of the leading-strand template. Solid lines represent fits to Eq. (2) in **b** and Eq. (3) in **c** and **d** (see Methods). **e** The ratio of fluorescence plateau intensity on Fork^dsLag to Fork^ssLag indicates that DNA stimulation of fork DNA unwinding by duplexing the lagging-strand arm is helicase dependent. All data represent mean ± SD ($n = 3$ independent experiments). Source data are provided as a Source Data file.

the CMG helicase, we analyzed unwinding kinetics of the same fork substrates by large T antigen and NS3, both 3′–5′ helicases. Although large T antigen hexamers are proposed to encircle dsDNA at the origin of replication[37], the helicase central channel within the motor domain is narrower than the cross section of dsDNA. As a result, large T antigen can unwind partial duplex DNA substrates lacking a 5′-ssDNA flap[38]. Because large T antigen should be less susceptible to sliding onto duplex DNA at the fork, stimulation of helicase activity by duplexing the 5′ tail of the fork is expected to be less-pronounced compared with CMG. Consistently, while fluorescence increase on Fork^dsLag was 10-fold higher than that of Fork^ssLag when unwound by CMG, this change was only 1.5-fold in large T antigen-catalyzed unwinding of the two substrates (Fig. 3c, e). This result suggests that regulation of DNA unwinding by replicative helicases through duplex DNA interaction at the fork correlates with their ability to translocate on dsDNA. To further test our model, we measured unwinding of fork DNA by NS3 from hepatitis C virus, a monomeric helicase lacking a central pore as found in replicative helicases[39,40]. As expected, no improvement of NS3 helicase activity was detected when the 5′ tail of the fork was duplexed

(Fig. 3d, e). Together, our data strongly suggest that duplexing the lagging-strand arm of fork DNA stimulates CMG helicase activity by preventing the helicase pore from encircling duplex DNA at the fork junction.

**Single-molecule analysis of CMG-driven fork unwinding**. To independently validate the results from fluorescence-based ensemble experiments demonstrated in Fig. 3, we devised a single-molecule assay to directly monitor DNA unwinding by CMG. To this end, fork DNA bearing an Atto647N fluorophore at the 5′-end of the displaced strand was immobilized on the glass surface of a microfluidic flow cell (Fig. 4a). CMG was introduced in the presence of ATPγS and allowed to bind the 3′ tail of surface-tethered fork DNA. Subsequently, ATP was drawn into the flow cell to initiate unwinding while the Atto647N-labeled strand was imaged with TIRF microscopy. Unwinding of the fork substrate was assessed from the loss of fluorescence signal upon dissociation of the displaced strand, which was coupled to the surface only through the biotin-modified translocation strand. After correcting for photobleaching (Supplementary Fig. 4), we found that 20–30% of surface-immobilized DNA was unwound

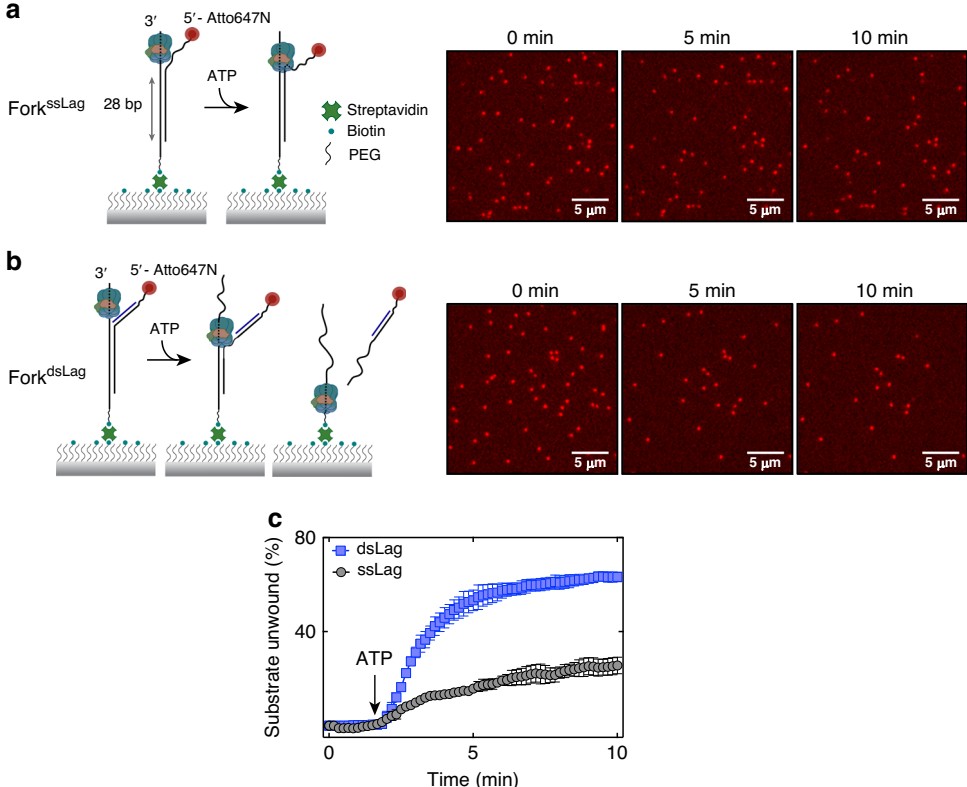

**Fig. 4 Single-molecule analysis of CMG-catalyzed fork unwinding. a, b** Visualizing CMG-driven unwinding of individual Atto647N-labeled surface-immobilized fork substrates with TIRF microscopy. Unwinding of 28-bp duplex region by CMG leads to dissociation of the fluorescent strand from the surface. Representative fields of view are shown at three time points on Fork$^{ssLag}$ **a** and Fork$^{dsLag}$ **b** following ATP addition ($n = 3$ independent experiments). **c** Percentage of molecules unwound as a function of time for Fork$^{ssLag}$ (gray, $N = 1215$ molecules from $n = 3$ independent experiments) and Fork$^{dsLag}$ (blue, $N = 1611$ molecules analyzed). Data represent mean ± SD from three independent experiments for each substrate. Source data are provided as a Source Data file.

by CMG (Fig. 4a, c, Fork$^{ssLag}$). To examine whether duplexing the lagging-strand arm of the fork alters the efficiency of unwinding, we annealed a complementary oligonucleotide to the 5′ ssDNA tail of immobilized fork DNA before CMG binding. Upon addition of ATP, 60–70% of Fork$^{dsLag}$ was unwound (Fig. 4b, c, Fork$^{dsLag}$). Therefore, single-molecule visualization of DNA unwinding confirms that duplexing the lagging-strand tail of fork DNA has a profound stimulatory effect on CMG helicase activity.

**RPA binding to the excluded strand promotes DNA unwinding.** At the eukaryotic replication fork, while the leading strand is proximately synthesized behind CMG, lagging-strand replication proceeds by discontinuous synthesis of Okazaki fragments. While the lagging-strand template near the fork junction remains single-stranded during fork progression, it is bound by the single-stranded binding protein, replication protein A (RPA), which may prevent CMG from partially encircling the excluded strand in its central pore. This model could explain how RPA speeds up dsDNA unwinding by CMG (Figs. 1 and 2)[7,17]. We tested whether RPA binding to the excluded strand can stimulate CMG's ability to unwind fork DNA using the fluorescence-quencher labeled DNA unwinding assay. After CMG was bound to fork DNA with ATPγS, RPA was added to bind the 5′ tail of the fork prior to ATP addition. A capture oligonucleotide was also included with ATP to sequester both excess CMG and RPA binding to DNA once unwinding began. Addition of 25 nM RPA led to threefold higher fluorescence signal on Fork$^{ssLag}$ (Fig. 5a). At this RPA concentration unwinding was strictly dependent on CMG (Supplementary Fig. 5a). Critically, RPA did not increase

unwinding of Fork$^{dsLag}$ by CMG (Fig. 5b), corroborating that RPA binding to the lagging-strand arm of Fork$^{ssLag}$ was responsible for the observed stimulation. These results suggest the binding of RPA to the lagging-strand template enhances CMG helicase activity by hindering engagement of the helicase with parental duplex at the fork junction providing mechanistic insight into how RPA speeds up CMG-driven DNA unwinding.

**Lagging-strand streptavidin enhances CMG helicase activity.** Our findings indicate that, when single-stranded, the excluded strand near the fork nexus is inhibitory to fork unwinding by CMG. Although one possible reason for the observed inhibition is the partial entrapment of the displaced strand in the helicase central pore, the other consideration is the interaction of the displaced strand with the outer surface of CMG. If a physical contact between the exterior of CMG and the lagging-strand template adversely affects helicase activity, as suggested for DnaB[29], hybridization of a complementary oligonucleotide or binding of RPA to the lagging-strand arm of a fork substrate may stimulate DNA unwinding by disrupting this interaction. To determine whether the inhibition of CMG helicase activity stems from its interaction with the displaced strand through the helicase exterior residues, we designed a fork DNA substrate containing a single biotin on the lagging-strand arm one nucleotide from the ssDNA–dsDNA junction. Although a streptavidin bound to biotin at this position should not prevent the external surface of the CMG from interacting with the lagging-strand arm, it is expected to prevent the helicase ring from encircling the excluded strand. Addition of streptavidin to fork DNA modified with

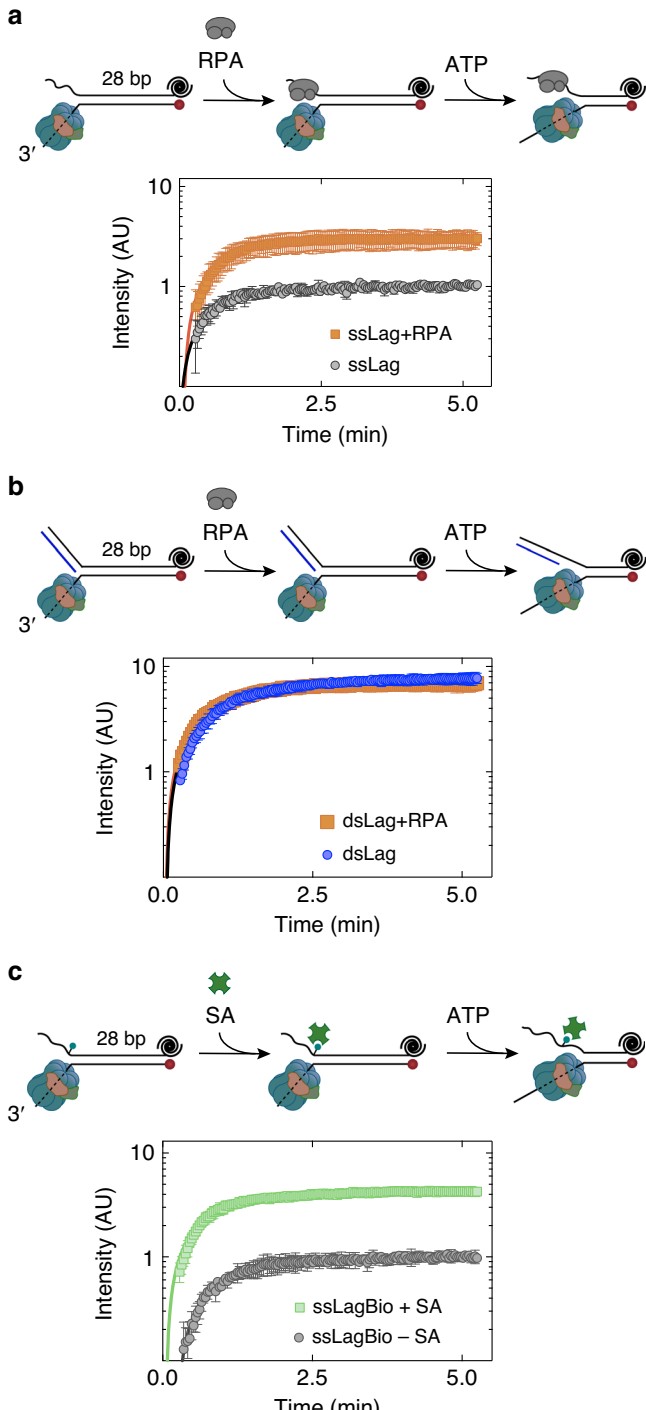

**Fig. 5 Binding of a protein to the excluded strand promotes DNA unwinding by CMG. a** Single-turnover time-course unwinding of Fork$^{ssLag}$ by CMG in the absence (black) and presence (orange) of RPA. **b** Single-turnover time-course unwinding of Fork$^{dsLag}$ by CMG in the absence (blue) and presence (orange) of RPA. **a**, **b** RPA was added after CMG binding to the fork. A competitor oligonucleotide was included with ATP addition to prevent further RPA and CMG binding during unwinding. Data represent mean ± SD ($n = 3$). **c** CMG-catalyzed single-turnover unwinding of fork DNA containing a single biotin on the lagging-strand arm in the absence (black) and presence (green) of streptavidin (SA). Data represent mean ± SD ($n = 3$ independent experiments). Solid lines are fits to Eq. (2) (Methods). Source data are provided as a Source Data file.

biotin on the 5′ tail enhanced CMG helicase activity (Fig. 5c) similar to the levels observed with duplexing the 5′ tail (Fig. 3b). Addition of streptavidin on a non-biotinylated fork substrate had no effect (Supplementary Fig. 5b, c), indicating that streptavidin promoted CMG-driven fork unwinding by binding to the 5′ tail of the fork template. Together, our data suggest that CMG pauses when engaged with the lagging-strand template via its central pore and that this non-productive helicase-fork arrangement is averted by binding of a protein to the lagging-strand template.

**DNA reannealing can free CMG from the duplex-engaged state.** It is conceivable that CMG occasionally stalls at the replication fork upon sliding onto duplex DNA when the lagging-strand template adjacent to the replication fork is not occupied by a protein such as RPA. To resume DNA unwinding, the helicase has to exit this paused state. As CMG exhibits a biased random walk during DNA unwinding[17], we reasoned that the helicase should be able to move backwards and revert to a strand exclusion mode if paused owing to encircling a short tract of dsDNA. To investigate whether CMG can exit a duplex-engaged mode, we allowed the helicase to first enter into the trapped state on the fork substrate used in Fig. 3 (Fork$^{ssLag}$) consisting of 28-bp dsDNA. To this end, CMG was first bound to Fork$^{ssLag}$ with ATPγS and its translocation was subsequently triggered with ATP. When an oligonucleotide complementary only to the single-stranded 5′ tail of the fork (Comp$^{LagTail}$) is added before (Fig. 6a, Comp$^{LagTail}$ → ATP, blue) or together with ATP (Supplementary Fig. 3e), DNA unwinding was markedly increased compared with the reaction lacking Comp$^{LagTail}$. Surprisingly, when introduced 2 minutes after ATP, Comp$^{LagTail}$ did not stimulate DNA unwinding (Fig. 6a, ATP → Comp$^{LagTail}$, red), suggesting that once CMG engages with duplex DNA at the fork nexus, the helicase is permanently stalled, unable to retract and disengage from the parental dsDNA. As the reverse motion of CMG was inferred from DNA rezipping in magnetic-tweezers measurements[17], we envisaged that fork reannealing ahead of CMG may be required to push the helicase backwards to exit a paused state. In this scenario, when CMG engages with the parental duplex after ATP addition on the fork substrate used in Fig. 6a, it cannot exit this paused state because the two non-complementary ssDNA tails does not anneal. As a result, adding Comp$^{LagTail}$ well after ATP does not rescue unwinding (Fig. 6a, ATP → Comp$^{LagTail}$, red).

To investigate whether CMG can exit a duplex-engaged mode owing to DNA reannealing, we measured its helicase activity on a fork substrate containing 60-bp dsDNA and d(GGCA)$_{10}$ 5′ tail that prevents the helicase entering the paused state near the ss-dsDNA junction (Supplementary Fig. 3a). The presence of relatively long dsDNA on this substrate makes it likely that CMG will enter the aforementioned stalled state while unwinding the 60-bp duplex section. However, if DNA reannealing can free CMG from the duplex-engaged mode, helicase pausing within the 60-bp region should not be permanent, as the two complementary strands in front of the helicase can reanneal (Fig. 6b). In this assay, we added a 32-nt oligonucleotide complementary to the excluded strand only within the parental duplex region (Comp$^{LagParent}$) either before or after ATP. When included before ATP, Comp$^{LagParent}$ enhanced unwinding by CMG (Fig. 6b, Comp$^{LagParent}$ → ATP, blue), suggesting that the helicase engages with dsDNA within the 60-bp parental duplex and stalls during unwinding. Strikingly, CMG-mediated unwinding of this substrate quickly recovered even when Comp$^{LagParent}$ was introduced 20 minutes after

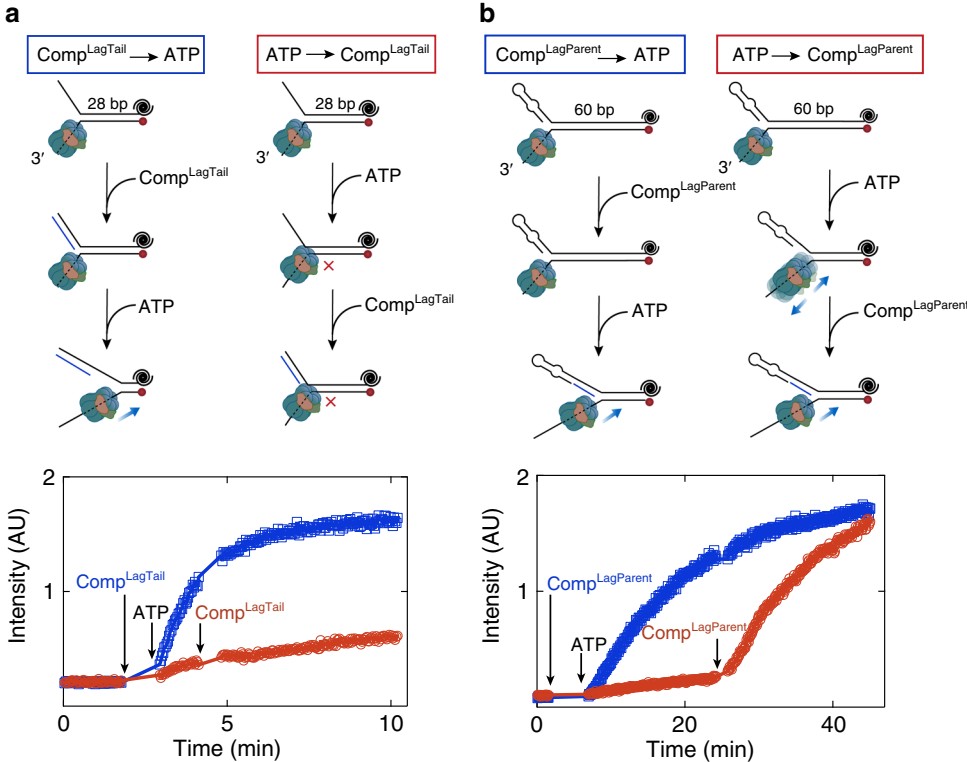

**Fig. 6 DNA reannealing can free CMG from the duplex-engaged state. a** Single-turnover unwinding of fork DNA with a single-stranded lagging-strand arm and 28 bp parental duplex. CMG was bound to the fork in the presence of ATPγS. Subsequently ATP was added and unwinding was monitored through Cy5 fluorescence. An oligonucleotide complementary to the lagging-strand arm (Comp$^{LagTail}$) was added either before (blue, Comp$^{LagTail}$→ATP) or 2 minutes after (red, ATP→Comp$^{LagTail}$) ATP. **b** Single-turnover unwinding of fork DNA with d(GGCA)$_{10}$ lagging-strand arm and 60-bp parental duplex. CMG was bound to the fork in the presence of ATPγS. Subsequently ATP was added and unwinding was monitored through Cy5 fluorescence. An oligonucleotide complementary to the lagging-strand template within the 60-bp parental dsDNA (Comp$^{LagParent}$) was added either before (blue, Comp$^{LagParent}$→ATP) or 20 minutes after (red, ATP→Comp$^{LagParent}$) ATP. Source data are provided as a Source Data file.

ATP (Fig. 6b, ATP → Comp$^{LagParent}$, red). To determine whether this stimulation is owing to the suppression of unwound DNA strands rehybridizing behind the helicase, we tested the impact of an oligonucleotide complementary to the leading-strand template within the parental duplex (Comp$^{LeadParent}$) on unwinding. The increase in DNA unwinding by Comp$^{LeadParent}$ was significantly lower than that by Comp$^{LagParent}$ (Supplementary Fig. 6), indicating that increased helicase activity by Comp$^{LagParent}$ cannot be solely ascribed to an inhibition of fork DNA reannealing in the wake of CMG. Together, the data are consistent with the notion that if the parental duplex invades into the CMG central channel, reannealing of DNA ahead of the helicase can liberate CMG from this inactive fork-binding mode by promoting backwards translocation.

**CMG bypasses a lagging-strand DPC without stalling.** Cryo-EM structures of CMG on fork DNA exhibited a short stretch of dsDNA enclosed by the N-terminal zinc finger (ZF) protrusions of Mcm2-7 (41–43). A possible exit route for the displaced lagging-strand template was proposed to be formed by cavities between MCM ZF protrusions[42,43]. If the displaced strand exits through a narrow opening between MCM ZF domains during unwinding, CMG is expected to pause when colliding with a bulky obstacle on this strand. We previously demonstrated that although a methyltransferase crosslinked to the lagging-strand template slowed down DNA unwinding by CMG, a covalent lagging-strand streptavidin block did not alter CMG dynamics[7]. There are two major differences between these two types of DNA-protein crosslinks (DPCs). First, methyltransferase interacts with both DNA strands and increases the

stability of duplex DNA, hence impeding CMG translocation[7]. Second, although methyltransferase was directly crosslinked to a base, streptavidin was attached to DNA through a flexible linker. Thus, the absence of CMG slowing down at a lagging-strand streptavidin barrier may be owing to the linker escaping through the narrow channel between MCM ZF domains. To test this possibility, we used 5-formylcytosine (5fC) modification to form a DPC lacking a linker between DNA and streptavidin[44,45] (Fig. 7a). We generated a fork substrate containing a single 5fC-streptavidin cross-link on the lagging-strand template in the middle of 60-bp parental duplex (Supplementary Fig. 7). We measured unwinding dynamics using the fluorescence-based single-turnover kinetic unwinding assay as described in Fig. 3. Importantly, we found that fork DNA containing a 5fC-streptavidin cross-link was unwound by CMG as efficiently and as rapidly as the non-adducted substrate (Fig. 7b). This result suggests that the lagging-strand template is unlikely to be expelled through a tight gap in the N-terminal region of Mcm2-7 during unwinding. Together with our data demonstrating that duplex engagement by CMG severely slows it down during DNA unwinding, we favor a model in which CMG engaged with the fork by encircling parental duplex through MCM ZF domains reflects a stalled helicase. Accordingly, multiple different conformations of parental duplex with respect to the N-terminal region of MCM were observed in *Drosophila* CMG/fork DNA structure suggesting plasticity in duplex engagement by the helicase[43].

**Discussion**
Here, to uncover the origin of poor helicase activity of isolated CMG, we studied how DNA unwinding by the helicase is

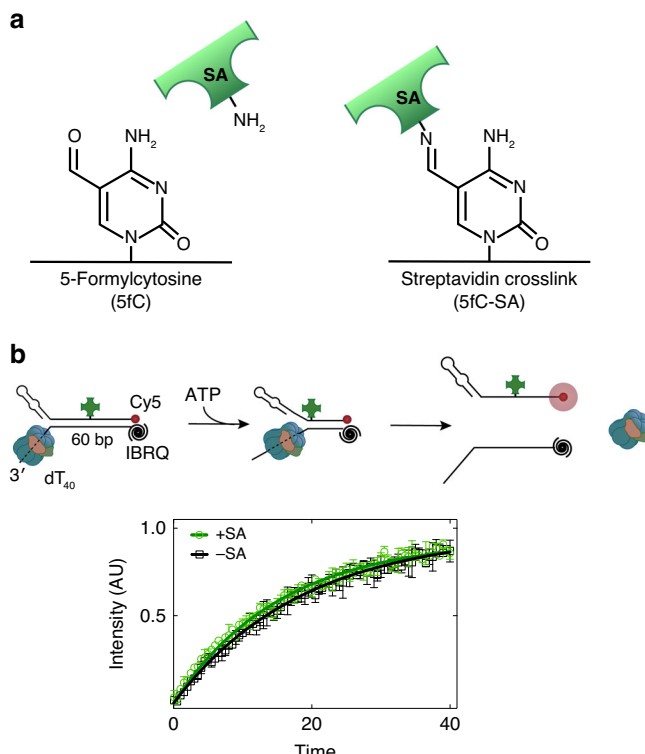

**Fig. 7 CMG bypasses a protein directly crosslinked to the excluded strand with no detectable stalling. a** 5-formyl-cytosine-modified oligonucleotide covalently crosslinks to streptavidin through primary amines. **b** Single-turnover CMG-mediated unwinding of fork DNA substrates with and without a 5fC-streptavidin cross-link. The 5′ tail of the DNA substrate contained d(GGCA)$_{10}$ sequence, which generates secondary structures and prevents helicase pausing at the fork junction. Fork DNA included 60-bp duplex region and were modified with Cy5 at the 3′ end of the lagging-strand template and Iowa Black RQ (IBRQ) dark quencher at the 5′ end of the leading-strand template. Separation of the complementary strands by CMG leads to fluorescence increase, a proxy for DNA unwinding. CMG unwinds 5fC-streptavidin-crosslinked and non-crosslinked substrates with the same kinetics. Data represent mean ± SD ($n = 3$ independent experiments). Solid lines represent fits to Eq. (2) (Methods). Source data are provided as a Source Data file.

regulated by its interaction with the replication fork. We found that interaction of CMG with ss-dsDNA fork junction leads to helicase stalling likely owing to partial entrapment of parental duplex within the helicase central channel, which explains the extremely slow translocation of CMG while unwinding the fork (Fig. 8a).

Similar to CMG, duplex unwinding by T7 gp4 and *E. coli* DnaB, which can translocate on dsDNA, is hindered by their interactions with duplex DNA at the fork junction[14,33,35]. Measurement of DNA hairpin unwinding by T4 gp41, T7 gp4, and *E. coli* DnaB with single-molecule magnetic and optical tweezers showed that these replicative helicases unwind dsDNA 5–10-fold slower compared with their ssDNA translocation rates[19,23,47]. Similarly, while CMG unwinds dsDNA at rates of 0.1–0.5 bp s$^{-1}$ (17), it translocates on ssDNA with an average rate of 5–10 nts$^{-1}$ [22], more than 10-fold variance. Large T antigen, another ring-shaped replicative helicase, unwinds dsDNA only twofold slower than it translocates on ssDNA[48]. Although large T antigen does not slide onto dsDNA upon meeting a flush ss-dsDNA junction, it may still interact with the parental duplex at the fork junction to some extent. This idea is in line with our observation that duplexing the

lagging-strand arm of fork DNA stimulates T antigen helicase activity, albeit to a much lesser degree than CMG (Fig. 3). Together, our results support a model whereby inhibition of DNA unwinding owing to parental duplex engagement is a shared characteristic among replicative helicases and suggests that the likelihood of helicase slowing down at the fork correlates with their capacity to translocate along dsDNA.

Coupling of the gp4 helicase with the leading-strand polymerase in the T7 replisome positions the helicase central channel axis perpendicular to the parental DNA[49]. This observation is in line with the prediction that the leading-strand polymerase stimulates DNA unwinding by gp4 by obstructing helicase engagement with duplex DNA. We speculate that faster fork rates seen in DnaB when coupled to the polymerase[24] may be through the same mechanism. Unlike bacteriophage and bacteria, the leading-strand polymerase, polymerase epsilon (Pol ε), is placed behind the helicase in eukaryotes. Thus, leading-strand synthesis would not hinder CMG invading onto parental DNA. On the contrary, Pol ε binding to DNA behind CMG may further lead to fork pausing because CMG backtracking appears to be essential to rescue the helicase from duplexed-engaged conformation (Fig. 6), which may explain CMG pausing at protein barriers in a Pol ε-dependent manner[50].

We demonstrate that binding of RPA to the lagging-strand arm of fork DNA prevents helicase engagement with the parental duplex near the fork junction (Fig. 8b). Therefore, we propose that binding of RPA to the lagging-strand template is essential for proper replication fork progression. Accordingly, we show that single CMG complexes can processively unwind thousands of base pairs of DNA in the presence of RPA at a rate similar to the speed of CMG on ssDNA[22]. Because RPA can diffuse along ssDNA[51], we envisage that new RPA from solution does not need to immediately bind to the emerging lagging-strand template near the helicase to support unwinding. We note that *E. coli* single-stranded binding protein also speeds up CMG-driven duplex unwinding[17]. Therefore, like RPA, other replication factors such as polymerase alpha-primase, Mcm10 and AND-1 (Ctf4), may also promote DNA unwinding by keeping the helicase from interacting with the parental duplex. The inability of CMG to drive extensive DNA unwinding in the absence of RPA should be beneficial for cells. Specifically, under conditions of RPA exhaustion, capture of the parental duplex by the N-terminal region of the MCM pore would lead to helicase slowing and prevent accumulation of unprotected ssDNA. When stalled owing to engagement with the parental dsDNA, we propose that reannealing of the parental duplex would drive CMG backwards and restore DNA unwinding (Fig. 8a). The plasticity of the eukaryotic replicative helicase to move backwards may be particularly important to keep the replisome in place during replication fork reversal[52] such that replication fork restart can take place without the need for new helicase loading onto DNA, which is inhibited in S phase[53]. In addition, 5′–3′ helicases may also help to rescue a duplex-engaged CMG by acting on the excluded strand at the replication fork. Mcm10, which is proposed to open a gate in the CMG, may also facilitate displacement of duplex DNA from the helicase central channel[22].

It is not clear whether the recent CMG structures in complex with fork DNA exhibiting dsDNA within the MCM ZF domains represent the helicase in an active fork unwinding state or a paused state[41,43]. Because CMG can unwind dsDNA much more efficiently with the help of RPA, a high-resolution structure of CMG while unwinding duplex DNA in the presence of RPA should further elucidate how the helicase engages with DNA in the eukaryotic replisome.

The rate of DNA unwinding by *Drosophila* CMG in the presence of RPA (4.5 bp s$^{-1}$, Fig. 1) is still an order of magnitude

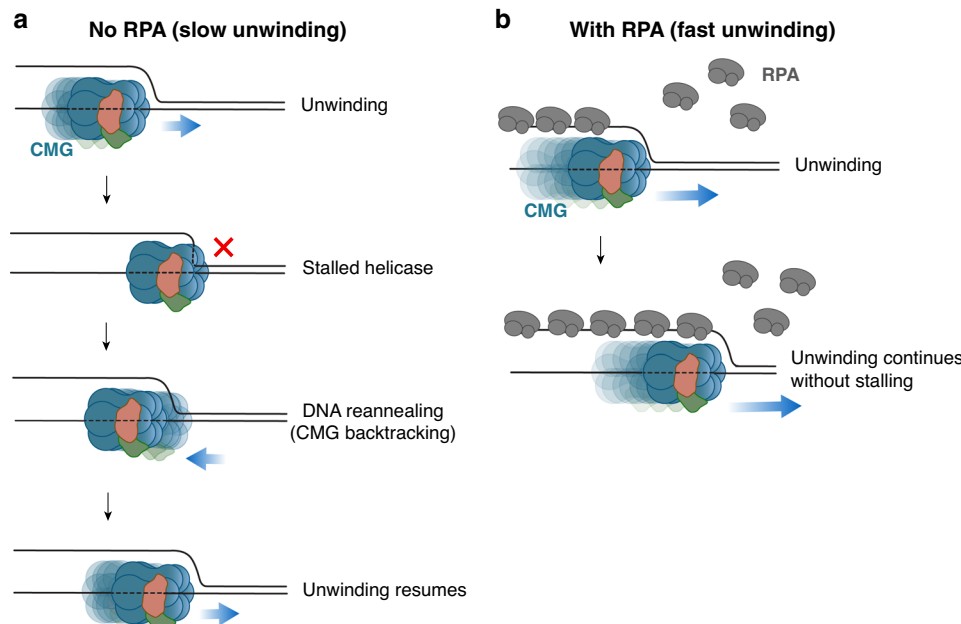

**Fig. 8 Model for RPA-facilitated DNA unwinding by CMG. a** While unwinding dsDNA, CMG often pauses owing to engagement with the parental duplex at the fork junction. DNA rezipping-induced CMG backtracking can rescue a stalled helicase. Owing to frequent pausing, CMG unwinds duplex DNA very slowly. **b** Binding of RPA to the lagging-strand template prevents CMG from engaging with duplex DNA at the fork junction. Thus, in the presence of RPA, CMG unwinds dsDNA at rates similar to its ssDNA translocation rate.

lower than average cellular replication fork rates in *Drosophila melanogaster* (~43 bp s$^{-1}$)[54]. The observation that unwinding rates are reduced when CMG uncouples from the leading-strand synthesis[26,27] indicates that pol ε increases replication fork rates. In yeast, Mrc1-Tof1-Csm3 complex (Claspin-Timeless-Tipin in metazoans) is reported to further enhance fork speed[46,55]. The molecular mechanism by which pol ε and Mrc1-Tof1-Csm3 speed up fork progression is unclear. It will be important to test whether these factors can stimulate DNA unwinding by CMG in the absence of DNA synthesis.

## Methods

**DNA substrates**. Supplementary Table 1 includes a list of oligonucleotides (oligos) used to prepare various DNA substrates and the corresponding figure numbers. The oligonucleotide sequences can be found in Supplementary Table 2.

To prepare fork DNA substrates, oligos (10 μM final), as listed in Supplementary Tables 1 and 2, were mixed in STE buffer (10 mM Tris-HCl pH 8.0, 100 mM NaCl, 1 mM ethylenediaminetetraacetic acid; EDTA), heated to 85 °C, and allowed to slowly cool down to room temperature. DNA substrates were ligated with T4 DNA Ligase (NEB) to seal nicks when necessary. The resulting products were separated on 8% polyacrylamide gel electrophoresis (PAGE) and purified by electroelution, unless stated otherwise. Electroelution was performed using 3.5 kDa MWCO dialysis tubing (Spectra/Por, Spectrum Labs) in 1× Tris/Borate/EDTA buffer.

Fork DNA containing 236-bp duplex and internal Cy5 (Fig. 2a) was made by PCR amplification of 314-bp fragment from pHY10[56] using primers Oligo-2 and Oligo-3 in the presence of Cy5-dCTP (GE Healthcare). The PCR reaction was purified using QIAquick PCR purification kit (Qiagen), digested with NcoI, and separated on 2% agarose gel. The resulting 211-bp fragment with 5′-CATG-3′ overhang was purified using gel extraction kit (Qiagen), ligated to 10-fold excess of short fork generated by annealing Oligo-4 and Oligo-5. The final product was separated on 8% PAGE and purified via electroelution. To make 252-bp duplex fork with downstream 28-bp duplex and 3′-Cy5-modified strand (Fig. 2b), 314-bp fragment from pHY10 was PCR amplified using primers Oligo-2 and Oligo-17. After purifying the reaction with QIAquick PCR purification kit, the substrate was cut with NcoI and HindIII (NEB), purified after separating on 2% agarose, and ligated to oligos that were annealed and purified from 8% PAGE. Whereas NcoI-digested end was ligated to Oligo-4/Oligo-5, HindIII-treated end was ligated to Oligo-15/Oligo-16/Oligo-Cy5-6. The final reaction was separated on 2.5% agarose and purified using electroelution. The substrate lacking the 40-nt 3′ polyT tail (Supplementary Fig. 2) was made using exactly the same protocol except Oligo-5 was replaced with Oligo-18.

To prepare 5fC-streptavidin-crosslinked fork substrate (Fig. 7), 1 nmol 5fC-modified oligo (Oligo-5fC) was incubated with 1 mg streptavidin (Sigma) in 100 μl phosphate-buffered saline (PBS) at 55 °C for 4 hours followed by 37 °C for 12 hours. DNA was separated on 8% PAGE in 1× TBE. The band corresponding to streptavidin-crosslinked oligo (1–2% of the reaction) was excised, purified via electroelution, annealed and ligated to other oligos as listed in Supplementary Table 1. The streptavidin-crosslinked fork was purified again by electroelution after separating on 8% PAGE.

Forked 10-kb linear DNA used in single-molecule assays (Fig. 1) was generated by treating λ DNA with ApaI (NEB). 10-kb fragment from ApaI-cut λ DNA was purified from a 0.5% agarose gel using Monarch gel extraction kit (NEB). We generated the fork end containing a 5′ biotin and a Cy3 on the 3′ dT$_{40}$ tail by annealing Oligo-Bio-4 and Oligo-Cy3-1, which leaves a 12-nt ssDNA complementary to one end of the 10 kb λ-ApaI fragment. To label the other end of 10 kb DNA with digoxigenin, a 0.5 kb fragment was PCR amplified from pUC19 vector with primers Oligo-1 and Oligo-8 in the presence of digoxigenin-11-dUTP (Roche). Digoxigenin-modified PCR substrate was then digested with ApaI and purified using QIAquick PCR purification kit. Subsequently, 2–3 μg of 10 kb DNA was mixed with 10-fold molar excess of Cy3-biotin-labeled fork substrate and digoxigenin-modified PCR template, and ligated with T4 DNA ligase. The reaction was mixed with 15 μg streptavidin for attachment to the biotin end of the fork, separated on 0.5% agarose in 1× TBE, and purified via electroelution.

## Protein expression and purification

*Drosophila melanogaster* CMG. *Drosophila* CMG was expressed, and purified as described before[7,10]. Each pFastBac1 (pFB) vector containing a single subunit of *Drosophila* CMG was transformed into DH10Bac *E. coli* competent cells (Thermo Fisher) to generate bacmids. Mcm3 subunit contained an N-terminal Flag tag for purification. Sf21 cells (10$^6$/ml) were used for the initial transfection (P1 stage), and in the subsequent virus amplification stage to make P2 stocks using Sf-900TM III SFM insect cell medium (Invitrogen/Gibco). In all virus amplification stages, cells were incubated at 27 °C, while shaking at 120 rpm. To further amplify virus stocks (P3 stage), 100 ml Sf9 cell cultures (0.5 × 10$^5$/ml) in Graces medium supplemented with 10% fetal calf serum for each subunit were infected with 0.5 ml P2 viruses and incubated in 500 ml Erlenmeyer sterile flasks (Corning) for ~100 hours. Total of 200 ml P3 viruses for each subunit were used to infect 4 L of Hi5 cells (10$^6$/ml) with a multiplicity of infection (MOI) of 5. Hi5 cells were divided into 500 ml aliquots using sterile 2 L roller bottles (Corning). After 48 hours, cells were harvested by centrifuging at 4500 × *g*. Cell pellets were first washed with PBS supplemented with 5 mM MgCl$_2$, resuspended in 200 ml Buffer C-Res (25 mM HEPES pH 7.5, 1 mM EDTA, 1 mM EGTA, 0.02% Tween-20, 10% glycerol, 15 mM KCl, 2 mM MgCl$_2$, 2 mM β-ME (2-mercaptoethanol), PI tablets) (50 ml buffer per 1 L Hi5 cell culture), and frozen in 10 ml aliquots on dry ice. Cell pellets were stored in −80 °C.

All purification steps were performed at 4 °C unless specified otherwise. Frozen cell pellets were thawed in lukewarm water, and lysed by 60–70 strokes using cell

homogenizer (Wheaton, 40 ml Dounce Tissue Grinder) on ice. Cell debris was removed by centrifugation at $24,000 \times g$ for 10 minutes. The supernatant was collected, and incubated with M2 agarose Flag beads (Sigma Aldrich) equilibrated with Buffer C (25 mM HEPES pH 7.5, 1 mM EDTA, 1 mM EGTA, 0.02% Tween-20, 10% glycerol, 1 mM dithiothreitol; DTT) for 2.5 hours. CMG was eluted from beads by incubating with Buffer C-100 (25 mM HEPES pH 7.5, 1 mM EDTA, 1 mM EGTA, 0.02% Tween-20, 10% glycerol, 100 mM KCl, 1 mM DTT) supplemented with 200 μg/ml peptide (DYKDDDDK) for 15 minutes at room temperature. The eluate was passed through 1 ml HiTrap SPFF column (GE Healthcare) equilibrated with Buffer C-100. CMG was separated with 100–550 mM KCl gradient using 5/50GL MonoQ column (GE Healthcare). Fractions containing CMG were pooled, diluted to 150 mM KCl, and loaded onto MonoQ PC 1.6/5GL (GE Healthcare) column equilibrated with Buffer C-150 (25 mM HEPES pH 7.5, 1 mM EDTA, 1 mM EGTA, 10% glycerol, 150 mM KCl, 1 mM DTT) to concentrate the sample. A gradient of 150–550 mM KCl was applied to elute CMG. Fractions containing CMG were pooled, and dialyzed against CMG-dialysis buffer (25 mM HEPES pH 7.5, 50 mM sodium acetate, 10 mM magnesium acetate, 10% glycerol, 1 mM DTT) for 2 hours.

To prepare the fluorescently-labeled CMG construct, site-directed mutagenesis (QuikChange, Agilent) was applied to pFB-Mcm3 vector to insert a TEV cleavage site (ENLYFQG) followed by four Gly residues downstream of the N-terminal Flag tag on Mcm3 for subsequent Sortase-mediated labeling. The construct was expressed and purified as before with slight modifications. After collected from 5/50GL MonoQ column, 1 ml of the sample was mixed with 50 μl TEV protease (1 mg/ml, EZCut TEV Protease, Biovision), and the mixture was dialyzed against Buffer C-100 overnight at 4 °C. TEV-treated sample was supplemented with 50 μM peptide NH$_2$-CHHHHHHHHHHLPETGG-COOH, labeled with LD655-MAL (Lumidyne Technologies) on the cysteine residue, 10 μg/ml Sortase enzyme and 5 mM CaCl$_2$, and incubated 30 minutes at 4 °C. Free peptide was removed by separating the sample through gel filtration (Superdex 200 Increase 10/300 GL) in Buffer C-100. The labeled construct was concentrated on MonoQ PC 1.6/5GL and dialyzed as described above.

*Sortase*. pET29-based expression vector for C-terminally 6xHis-tagged Sortase A pentamutant (Sortase P94R/D160N/D165A/K190E/K196T)[57] was obtained from Dr. David Liu (Harvard University). *E. coli* BL21(DE3) competent cells transformed with the expression vector were grown at 37 °C in lysogeny broth (LB) with 50 μg/ml kanamycin until OD = 0.5–0.8 and induced with 0.4 mM isopropyl β-D-1-thiogalactopyranoside (IPTG) for 3 hours at 30 °C. Cells were harvested by centrifugation and resuspended in Sortase-lysis buffer (50 mM Tris pH 8.0, 300 mM NaCl, 1 mM MgCl$_2$, 2 units/ml DNAseI (NEB), 260 nM aprotinin, 1.2 μM leupeptin, 1 mM phenylmethylsulfonyl fluoride). After lysing cells by sonication, cells were centrifuged at $7500 \times g$ for 20 minutes. The supernatant was supplemented with 10 mM imidazole, mixed with Ni-NTA agarose (1 ml bed volume pre-washed with Sortase-lysis buffer per 8 ml lysate), and incubated for 1 hour on a rotary shaker. The lysate-Ni-NTA mixture was loaded into a gravity column and washed twice with five bed volumes of Sortase-wash buffer (50 mM Tris pH 8.0, 300 mM NaCl, 1 mM MgCl$_2$, 20 mM imidazole). The protein was eluted from Ni-NTA beads in four rounds, each with one bed volume of Sortase-elution buffer (50 mM Tris pH 8.0, 300 mM NaCl, 1 mM MgCl$_2$, 250 mM imidazole). Fractions containing Sortase were combined and dialyzed against Sortase-storage buffer (25 mM Tris pH 7.0, 150 mM NaCl, 10% glycerol).

*Large T antigen*. Recombinant SV40 large T antigen was made using the baculovirus expression system and purified using a monoclonal antibody as described previously[16]. pFB with the full-length large T antigen was used to make the baculovirus. Sf21 cells maintained in SF-900-III were infected with the virus ($2 \times 10^8$ pfu/ml) using an MOI of 0.1 for 72 hours. Cell pellets were resuspended in 10 pellet volumes of L-Tag-resuspension buffer (20 mM Tris pH 9.0, 300 mM NaCl, 1 mM EDTA, 10% glycerol, 0.5% NP-40, 0.1 mM DTT) and incubated on ice for 15 minutes. The suspension was centrifuged at $25,000 \times g$ for 15 minutes. In all, 0.5 volume of L-Tag-neutralization buffer (100 mM Tris pH 6.8, 300 mM NaCl, 1 mM EDTA, 10% glycerol, 0.5% NP-40, 0.1 mM DTT) was added and mixed.

The protein was affinity purified using a mouse monoclonal antibody (PAb419). Antibody was coupled to Protein A sepharose beads (5 mg/ml) in PBS, incubated rotating overnight at 4 °C. Beads were washed with 15 ml 0.1 M sodium borate buffer, pH 9.0, and resuspended in 2 ml of the same buffer. To cross-link antibody to beads, 20 ml of 12.5 mg/ml dimethyl pimelimidate (Sigma) was mixed with the beads and incubated at room temperature rotating for 1 hour. Coupling reaction was quenched by incubating beads in 0.2 M ethanolamine pH 8.0, rotating for 1 hour at room temperature. Large T antigen suspended in neutralization buffer was first loaded onto protein A-only column, equilibrated with L-Tag-loading buffer (20 mM Tris pH 8.0, 300 mM NaCl, 1 mM EDTA, 10% glycerol, 0.5% NP-40, 0.1 mM DTT). Flow through was then loaded onto PAb419-conjugated column equilibrated with L-Tag-loading buffer. The column was washed with 50 ml L-Tag-loading buffer, then 50 ml L-Tag-wash buffer (50 mM Tris, pH 8.0, 1 M NaCl, 1 mM EDTA, 10% glycerol), followed by 20 ml L-Tag-EG buffer (50 mM Tris, pH 8.5, 500 mM NaCl, 1 mM EDTA, 10% glycerol, 10% ethylene glycol). Large T antigen was eluted with 5–10 ml of L-Tag-elution buffer (50 mM Tris, pH 8.5, 1 M NaCl, 10 mM MgCl$_2$, 1 mM EDTA, 10% glycerol, 55% ethylene glycol) and

dialyzed overnight into L-Tag-dialysis buffer (20 mM Tris pH 8.0, 10 mM NaCl, 1 mM EDTA, 50% glycerol, 1 mM DTT).

*NS3 helicase*. The expression plasmid containing N-terminal His-SUMO-tagged helicase domain of NS3 (SUMO-NS3h) gene was obtained from Charles Rice (Rockefeller University). NS3h was expressed, and purified as described previously[40]. NS3h plasmid was first transformed into Rosetta2 (DE3) competent cells (Novagen). Cells were grown in LB complemented with 50 μg/ml kanamycin and 34 μg/ml chloramphenicol at 37 °C. Shaking was interrupted at OD = 0.6, and the culture was kept at 4 °C for 30 minutes before induction. Cells were induced with 0.4 mM IPTG, transferred to a 16 °C incubator and left for 20 hours shaking at 240 rpm. Cells were harvested by centrifugation at $7500 \times g$ for 10 minutes. Pellets were kept in −80 °C.

To purify SUMO-NS3h, cell pellets were resuspended in 100 ml NS3-lysis buffer (20 mM Tris-HCl pH 8.5, 500 mM NaCl, 1 mM β-ME, 10 mM imidazole, 50 μg/ml lysozyme, 2.5 μg/ml RNase A) and incubated on ice for 20 minutes. Next, cells were stirred for 30 minutes at 4 °C and subsequently sonicated (3 seconds on, 10 seconds off, 20 cycles). Cell debris was removed by centrifugation at $24,000 \times g$ for 30 minutes. The supernatant was mixed and incubated with 3 ml Ni-NTA agarose beads (Qiagen) for 1 hour. The mixture was transferred to a 20-ml disposable column (Bio-Rad) to collect beads. The column was washed with NS3-wash buffer (20 mM Tris-HCl pH 8.5, 500 mM NaCl, 20 mM imidazole, 1 mM β-ME). SUMO-NS3h was eluted with NS3-Elution buffer (20 mM Tris-HCl pH 8.5, 150 mM NaCl, 250 mM imidazole, 1 mM β-ME). The eluate was loaded onto 5 ml HiTrap Q HP column (GE Healthcare) equilibrated with NS3 Buffer A (20 mM Tris-HCl pH 8.0, 100 mM NaCl, 1 mM β-ME). SUMO-NS3h was separated using a 100–500 mM NaCl gradient. Fractions containing SUMO-NS3h was pooled and concentrated to ~2.5 mg/ml using 30 kDa MWCO spin concentrator (Vivaspin).

To cleave the SUMO tag, SUMO-NS3h was diluted with NS3 Buffer D (20 mM Tris-HCl pH 8.0, 50 mM NaCl, 20% glycerol, 1 mM β-ME) to obtain 100 mM NaCl. 40 units of His-tagged SUMO protease (Invitrogen) was mixed with 1 mg SUMO-NS3h and incubated for 3 hours at 4 °C. The sample was then loaded onto 1 ml HisTrap FF column (GE Healthcare) to remove cleaved SUMO tag and SUMO protease. The flow-through which contained cleaved NS3h was collected and concentrated to 0.25 mg/ml using spin concentrator (Vivaspin, 30 kDa MWCO).

*Human RPA*. The expression plasmids for 6xHis-tagged non-fluorescent and EGFP-fused human RPA were obtained from Mauro Modesti (Cancer Research Center of Marseille, CNRS). Expression and purification of both RPA constructs were performed as described previously[58]. The plasmid was transformed into Rosetta/pLysS competent cells (Novagen). Cells were grown in LB media supplemented with 100 μg/ml ampicillin and 34 μg/ml chloramphenicol at 37 °C. At OD = 0.5, the temperature was reduced to 15 °C, RPA expression was induced with 1 mM IPTG, cells were further incubated for 20 hours. Cells were then harvested by centrifugation at $3500 \times g$ for 30 minutes. Supernatant was removed, and the cell pellet was washed with PBS. Pellets were stored in −80 °C.

To purify RPA, cell pellets were thawed in lukewarm water and resuspended in RPA-lysis buffer A (40 mM Tris-HCl pH 7.5, 1 M NaCl, 20% glycerol, 4 mM β-ME, 10 mM imidazole) supplemented with EDTA-free PI tablets. The suspended pellet was sonicated (3 seconds on, 10 seconds off, 20 cycles) to break the cells, and cell debris was removed by centrifugation at $20,000 \times g$ for 1 hour. Supernatant was filtered through 0.45 μm syringe filters (Millipore) and loaded onto 1 ml HisTrap FF (GE Healthcare) column equilibrated with RPA-lysis buffer B (20 mM Tris-HCl pH 7.5, 500 mM NaCl, 2 mM β-ME, 20% glycerol, 10 mM imidazole, 1 mM DTT). RPA was eluted by applying linear gradient of 10–300 mM imidazole. Fractions containing RPA were pooled and dialyzed against RPA-dialysis buffer A (20 mM Tris-HCl pH 7.5, 50 mM KCl, 0.5 mM EDTA, 10% glycerol, 1 mM DTT) using 3.5 kDa MWCO dialysis tubing (Generon) overnight. Dialyzed sample was filtered using 0.22 μm syringe filter (Millipore) as precipitation occurred. The sample was loaded onto HiTrap Heparin column (GE Healthcare) equilibrated with RPA-dialysis buffer A. The protein was eluted by applying 50–500 mM KCl gradient. Fractions containing RPA were pooled and dialyzed into RPA-dialysis buffer B (20 mM Tris-HCl pH 7.5, 50 mM KCl, 0.5 mM EDTA, 25% glycerol, 1 mM DTT) using 3.5 kDa MWCO dialysis tubing (Generon) for 2 hours.

**Gel-based DNA-unwinding assays**. To analyze unwinding of Cy5-labeled fork DNA containing 50-bp duplex region (Supplementary Fig. 3a), 3 nM DNA was incubated with 50 nM CMG in CMG-binding buffer (25 mM HEPES pH 7.5, 5 mM NaCl, 10 mM magnesium acetate, 5 mM DTT, 0.1 mg/ml bovine serum albumin) supplemented with 0.1 mM ATPγS at 37 °C for 1–2 hours in a 6 μl total volume. In all, 6 μl of ATP mix (CMG-binding buffer supplemented with 5 mM ATP) was added to initiate unwinding and samples were incubated at 30 °C for further 10 minutes. Reactions were terminated by addition of 3 μl stop buffer (0.5% SDS, 20 mM EDTA). To prevent aggregation of CMG-bound DNA that results in some DNA being stuck in the well during electrophoresis, 25 μM of 40-nt polyT oligo (Oligo-6) was included in the stop buffer. Reactions were separated on 8% PAGE in 1× TBE.

To measure unwinding of internally Cy5-modified 236 bp long DNA substrate (Figs. 2a), 5 nM fork substrate was incubated with 20 nM CMG in CMG-binding

buffer supplemented with 0.1 mM ATPγS at 37 °C for 1–2 hours. ATP and RPA was added to final concentrations of 2.5 mM and 150 nM, respectively. After incubating for 1 hour at 30 °C, the reactions were stopped with 0.1% SDS, and separated on 3% agarose in 1× TBE supplemented with 0.1% SDS. DNA unwinding assays with 252-bp duplex fork and downstream Cy5-modified strand (Fig. 2b) were performed similarly. When RPA was omitted from ATP-containing buffer (Fig. 2c), oligonucleotides Oligo-6 (to capture free CMG) and Oligo-21 (to prevent Cy5 strand reannealing to long DNA) were added each at a final concentration of 250 nM. The reactions were incubated at 30 °C for indicated times, quenched by adding 0.1% SDS. The stop solution for reactions containing RPA was also supplemented with Oligo-21 (250 nM final) to prevent reannealing of Cy5-modified strand upon RPA dissociation. Finally, DNA was separated on 8% PAGE in 1× TBE supplemented with 0.1% SDS. To quantify CMG-dependent unwinding in the presence of RPA, the data was corrected for Cy5-modified strand displaced by RPA alone.

Gels were imaged on Fujifilm, SLA-5000 scanner using 635-nm laser and Fujifilm LPR/R665 filter. Band intensities were quantified using ImageJ.

**Real-time DNA unwinding measurements on plate reader.** In all, 5 nM of Cy5/quencher dual-labeled fork substrates were incubated with 50 nM CMG in CMG-binding buffer in the presence of 0.1 mM ATPγS at 37 °C for 1–2 hours. To avoid non-specific binding of CMG and DNA to microplate wells (Nunc 384 shallow well plate, black, 264705), wells were pre-blocked by incubation with CMG-binding buffer supplemented with 1 mg/ml bovine serum albumin (BSA) for at least 30 minutes. In all, 5 μl of CMG-DNA mixture was transferred to a well, and 15 μl of ATP mix (CMG-binding buffer containing 3.3 mM ATP) supplemented with 1.5 μM 40-nt polyT oligo (Oligo-6) was added to the reaction. Unwinding assays performed with large T antigen were essentially the same and contained 0.1 mg/ml large T antigen in DNA/helicase binding reaction. For unwinding assays with NS3h, 5 nM of DNA substrate was mixed with 200 nM of NS3h in NS3-MOPS buffer (20 mM MOPS-NaOH pH 6.5, 30 mM NaCl, 3 mM MgCl₂, 1% glycerol, 0.2% Triton X-100, 1 mM DTT), incubated for 1 hour at 37 °C. In all, 5 μl of NS3h/DNA mix was transferred to a well and 15 μl of ATP mix (3.3 mM ATP in NS3-MOPS buffer) supplemented with 1.5 μM 40-nt polyT oligo (Oligo-6) was added start unwinding. When measuring unwinding of fork DNA that contains 28 bp duplex (Figs. 3, 5, and 6a), ATP mix was also supplemented with 0.5 μM Oligo-13 to prevent reannealing of the Cy5-labeled strand to the quencher-modified strand. Comp$^{\text{LagTail}}$ used for the fork substrate containing 28-bp duplex region (Figs. 3e and 6a) was Oligo-12. Comp$^{\text{LagParent}}$ and Comp$^{\text{LeadParent}}$ used in Fig. 6b and Supplementary Fig. 6 for the 60-bp fork DNA were Oligo-20 and Oligo-22, respectively.

Cy5 fluorescence intensity was recorded on a PHERAstar FS (BMG Labtech) with excitation and emission wavelengths of 640 and 680 nm, respectively. Data was collected with 2- or 5-seconds intervals with 10 flashes/measurement at 25 °C. Measured signals were normalized using Prism 7, and plotted as a function of time.

**Single-molecule DNA unwinding assays.** Microfluidic flow cells used in single-molecule assays were made by sandwiching double-sided tape (TESA SE, TESA 4965) between a coverslip, which was coated with PEG and PEG-biotin (Laysan Bio), and a non-functionalized glass slide as described in detail in ref. [59].

*Unwinding of Atto647N-labeled short fork substrates.* For single-molecule analysis of Atto647N-labeled fork DNA, coverslip surface was first coated with streptavidin by drawing 0.2 mg/ml streptavidin in PBS into the microfluidic flow chamber using a syringe pump (Harvard Apparatus), and incubating for 20 minutes. The flow chamber was extensively washed with blocking buffer (20 mM Tris-HCl pH 7.5, 50 mM NaCl, 2 mM EDTA, 0.2 mg/ml BSA) to remove excess streptavidin. The channel was washed with DNA-dilution buffer (20 mM Tris-HCl pH 7.5, 50 mM NaCl, 2 mM MgCl₂, 2 mM EDTA, 0.05 mg/ml BSA). Subsequently, 10 pM Atto647N-labeled fork DNA (Fork$^{\text{ssLag}}$-TIRF) containing biotin at one end was introduced in DNA-dilution buffer and incubated for 2–3 min for binding to the surface. The flow channel was washed with DNA-dilution buffer to remove unbound DNA molecules. In all, 40 nM CMG in 30 μl of TIRF-CMG-loading buffer A (10 mM Tris-HCl pH 7.5, 12 mM MgCl₂, 15 mM NaCl, 10% glycerol, 0.8 mg/ml BSA, 12.5 mM DTT, 0.3 mM ATPγS) was drawn into the chamber, and incubated for 1 hour. To reduce photobleaching, the flow channel was washed with Atto-imaging buffer (10 mM Tris-HCl pH 7.5, 12 mM MgCl₂, 15 mM NaCl, 10% glycerol, 0.8 mg/ml BSA, 12.5 mM DTT, 1% glucose, 0.02 mg/ml glucose oxidase (Sigma), 0.04 mg/ml catalase (Sigma)) supplemented with 0.3 mM ATPγS. Surface-tethered DNA was imaged using 647-nm laser at 10-second intervals. After imaging 80 seconds in ATPγS-containing buffer, unwinding was initiated by drawing Atto-imaging buffer containing 3.3 mM ATP into the channel while continuing to collect images. To perform unwinding assays with Fork$^{\text{dsLag}}$ substrate, after immobilizing Fork$^{\text{ssLag}}$ on the coverslip surface, 50 nM Oligo-12 was introduced in DNA-dilution buffer and incubated for 10 minutes. Flow cell was then washed with DNA-dilution buffer to remove excess oligo before introducing CMG.

*Unwinding of 10-kb long stretched DNA.* In all, 15 pM of 10-kb DNA bound to streptavidin on the forked end was introduced in blocking buffer into a flow cell

with polyethylene glycol (PEG)-biotin-functionalized glass surface and incubated for 1 hour for surface binding. The flow cell was washed with blocking buffer to remove free DNA. Anti-digoxigenin- (anti-dig) coated microspheres (0.05% w/v in blocking buffer) were introduced and incubated for 1 hour for binding to the free digoxigenin-modified end of surface-immobilized DNA molecules. Excess beads were removed by washing the flow cell with blocking buffer. To stretch DNA and attach anti-dig-conjugated beads to the surface, digoxigenin-modified streptavidin (dig-streptavidin, 3 μg/ml) was drawn at 0.2 ml/min in blocking buffer. Free dig-streptavidin was washed out by flushing the flow cell with blocking buffer. CMG-blocking buffer (CMG-binding buffer containing 0.8 mg/ml casein (Sigma Aldrich)) supplemented with 0.3 mM ATPγS was drawn into the flow cell and incubated for 10 minutes. LD655-labeled CMG (15 nM final in CMG-blocking buffer supplemented with 0.3 mM ATPγS) was introduced and incubated for 1 hour. To remove free CMG and initiate DNA unwinding, 20 nM EGFP-RPA was drawn in CMG-blocking buffer containing 3.3 mM ATP, 1% glucose, 0.02 mg/ml glucose oxidase, and 0.04 mg/ml catalase.

*Preparation of anti-digoxigenin-conjugated microspheres.* In all, 100 μl 5% w/v of 0.45 μm-diameter carboxyl-functionalized polystyrene microspheres (Spherotech, CP-05-10) were washed twice with 0.5 ml coupling buffer (sodium acetate pH 5.0) by centrifugation and resuspended in 0.8 ml coupling buffer. In total, 0.2 mg polyclonal anti-dig antibody (Roche, 11333089001) and 1 mg BSA were dissolved in 0.2 ml coupling buffer and mixed into carboxylated beads. The mixture was incubated 3 hours at room temperature on a rotator. Excess anti-dig and BSA were removed by washing microspheres with 1 ml PBS three consecutive rounds by centrifugation. Microspheres were resuspended in 0.2 ml PBS and stored at 4 °C. Before introducing into a flow cell, 2 μl anti-dig microspheres were washed once with 100 μl blocking buffer, resuspended in 100 μl blocking buffer, and briefly sonicated in an ultrasonic water bath (VWR) to break aggregates.

*Preparation of streptavidin-digoxigenin conjugate.* In total, 10 mg streptavidin (Sigma) was dissolved in sodium bicarbonate pH 8.2. 1 mg digoxigenin-NHS (Sigma, 55865) was dissolved in 100 μl dimethyl sulfoxide, mixed into streptavidin solution, and incubated 3 hours at room temperature rotating. Unconjugated digoxenin was removed by exchanging buffer five times with 10 ml PBS each round using a spin concentrator (Vivaspin, 20 ml, 30 kDa MWCO). Streptavidin-digoxigenin conjugate was finally concentrated to 3 mg/ml and stored at 4 °C.

*Microscope setup and image acquisition.* DNA-unwinding assays performed with immobilized DNA molecules were imaged on an objective-type TIRF configuration using an inverted microscope (Ti-E, Nikon) equipped with a ×100 oil objective (HP Apo TIRF 100xH, N.A. = 1.49, Nikon) and automated focus. Fluorescence of Atto647N and LD655 were recorded with excitation wavelength of 647 nm, whereas Cy3 and EGFP were illuminated with 561-nm and 488-nm lasers, respectively. Images were collected at 100–200 ms exposures per frame on an Andor iXon 897 back-illuminated electron-multiplying CCD camera (Andor Technology).

**Data analysis.** Images collected during single-molecule experiments were analyzed using NIS-Elements software (Nikon). Images were first aligned to correct the effect of stage drift over time. Bright spots corresponding to Atto647-dye conjugated DNA molecules above a custom threshold observed in the first frame were selected. The fluorescence intensity of each spot at each frame were measured, and exported for further analysis. Reported unwinding efficiencies observed in Fork$^{\text{ssLag}}$ and Fork$^{\text{dsLag}}$ were measured by dividing the cumulative number molecules that disappeared (i.e., molecules fully unwound) by the total number of molecules present on the first frame. Rate of DNA unwinding on 10-kb linear DNA substrates was measured through the growth rate of EGFP-RPA tracts.

**Fitting and normalization.** Fluorescence intensity values on plate reader assays were normalized by fitting the data to Eq. (1)[60] for integer values of $m$:

$$f_{ss}(t) = 1 - \sum_{r=1}^{m} \frac{k_{obs} t^{(r-1)}}{(r-1)!} e^{-k_{obs} t} \qquad (1)$$

where $f_{ss}(t)$ is time-dependent extent of DNA unwinding, m is the number of steps, $k_{obs}$ is the observed unwinding rate and $t$ is time. Fluorescence-time traces from CMG unwinding assays were fit using Eq. (2) ($m = 1$).

$$f_{ss}(t) = 1 - e^{-k_{obs} t} \qquad (2)$$

The data from large T antigen- and NS3-mediated DNA-unwinding assays were fitted with using Eq. (3) ($m = 2$) owing to the presence of a lag phase at early time points.

$$f_{ss}(t) = 1 - (1 + k_{obs} t) e^{-k_{obs} t} \qquad (3)$$

Constant values obtained subsequent to fitting were used to normalize unwinding signals. Normalized data were averaged and plotted against time.

**Statistical analysis**. Throughout the manuscript, the data are represented as average ± standard deviation of multiple experiments. Prism (GraphPad Software, La Jolla, CA, USA) was used to plot all graphs presented and for statistical analysis in this study. ImageJ was used to quantify band intensities in gel images.

**Reporting summary**. Further information on research design is available in the Nature Research Reporting Summary linked to this article.

## Data availability

Data supporting the findings of this study are available from the corresponding author upon reasonable request. Source data are provided with this paper.

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

## Acknowledgements

We thank Daniel Burnham and Alessandro Costa for critical reading of the manuscript. Additional thanks to Mauro Modesti for RPA and EGFP-RPA expression vectors, Charles Rice for NS3 expression vector, and David Liu for Sortase expression vector. Peptide Chemistry and Cell Services science technology platforms at the Francis Crick Institute provided peptides. This work was supported by the Francis Crick Institute, which receives its core funding from Cancer Research UK (FC001221), the UK Medical Research Council (FC001221), and the Wellcome Trust (FC001221). G.C. was supported by a PhD fellowship from the Boehringer Ingelheim Fonds.

## Author contributions

H.B.K and H.Y. designed the study and conducted experiments. S.X. and G.C. expressed and purified LD655-labeled CMG. M.S.S. expressed and purified Sortase. H.B.K. purified all other recombinant proteins. H.B.K and H.Y. interpreted the data and wrote the paper with input from other authors.

## Competing interests

The authors declare no competing interests.
