## [Peer Review File · Nature Communications]

Reviewers' comments:

Reviewer #1 (Remarks to the Author):

Review of "Duplex DNA engagement and a replisome factor oppositely regulate the speed of a replicative DNA helicase" by Kose et al. The authors have investigated the interaction of the eukaryotic CMG helicase with both strands of DNA as mediated by the single stranded binding protein, RPA, using both bulk and single molecule assays. They provide a plausible explanation of the observed stalling when naked ssDNA is the excluded strand that suggests duplex DNA enters the central pore entering a nonproductive state. RPA binding (or hairpin DNA) to the excluded strand prevents this pause and stimulates unwinding. The experimental quality is excellent, however there are few additional experiments that are needed to better support their conclusions and eliminate alternative hypotheses.

- 1) Figure 1- An important control would be to monitor CMG unwinding in the absence of RPA using the same experimental setup to directly compare the rates of unwinding.
- 2) Line 157- "extensive" unwinding is stated with RPA, but Figure 2a, lane 4, shows less than 50% unwound. A better way to state this would be 'increased' unwinding compared to lane 3? (also number of the lanes in Figure 2a, c, and d would help)
- 3) A control is needed to show that RPA cannot displace the stable hairpin and bind resulting ssDNA that is routinely used on the lagging strand.
- 4) In Figure 2d, could competitor DNA complementary to the translocating strand increase the unwinding rate as well. Or is it specifically dependent on RPA.
- 5) That said, could other SSBs (gp32 or SSB) increase the unwinding rate similar to RPA or is it species dependent.
- 6) Moreover, in later Figures 3-5, you show that RPA increases the initiation of unwinding which can lead to an increased rate of unwinding (Fig 5). Therefore, experiments in Figure 2 should be repeated with RPA bound prior to initiation with CMG to compare directly with Fig 2c.
- 7) Figure 4a, it seems possible that 50% of the CMG molecules also bind the 5' arm, thus rendering them inactive for unwinding. That is the reason you see a 2-fold decrease in unwound product compared to dsLag. This could also explain the results in Figure 5a where there is a 3-fold difference in fluorescence intensity.
- 8) Figure 4c, it is not clear where exactly the biotin is incorporated on the ssLag. Is it in a position to prevent your model of ds engagement for stalling? It would be useful to vary the position of the biotin (on the Lag strand, at the duplex junction, and then several bases into the duplex) to test this model. Need to expand and test other biotin locations in combination with Figure 7.
- 9) Figure 5a, Why doesn't the addition of the Complag DNA restore unwinding as in 5b?
- 10) And in this experiment 5a, is the Complag only complementary to the 5' arm or parts of the duplex region as well? This is not clear based on your discussion or in the Figure. At the bottom of the blue scheme in 5a, it appears only complementary to the 5' arm. This appears different than in 5b, where DNALag has complementary to the duplex excluded strand.
- 11) Points 9 and 10 above are important to understand prior to the interpretation of the results as it may explain the differences in the red traces for both. It is frustrating that the authors did not better explain this important facet of the experiment.
- 12) Figure 7 may be useful in exploiting SA to distinguish between duplex sliding and excluded strand engagement, however, as is, it is not complete. There is no difference in the unwinding rates +/-SA, but this substrate (ssDNA 5' tail) is the one that should show slow unwinding. This experiment needs 1) repeat with either duplex 5'-tail or hairpin, 2) addition of complementary DNALag and 3) addition of and effect of RPA as well as some combination of all the 3 points.
- 13) I am not sure any of your experiments can absolutely eliminate the possibility that the ssLag can interact with the exterior of the CMG and slow unwinding. If the SA experiments (Fig 5c) are designed correctly, then this may be good support, however, it is not clear that the biotin is in the correct position to absolutely support your conclusion.
- 14) There are no direct measurements or monitoring of the duplex DNA region during a purported

stalled state. All experiments are indirect and could be explained by an excluded strand interaction.

15) Figure 8: this cartoon is used to explain your model, but nowhere did you test the RPA exhaustion model. You would need to test unwinding under different RPA concentrations or pulse chase experiments. Otherwise, Figure 8 should stick to the idea of duplex sliding/stalling, and/or whether you can fully eliminate excluded strand interaction.

Minor Points

1) Figure supplement 1c, there appears to be a second Cy3 red star on the DNA in the bottom panel of the cartoon which shouldn't be there.

2) MCM10 and CTF4 are mentioned exactly once in the introduction. Based on 2 recent papers from the O'Donnell group they may also have a role in preventing or regulating duplex sliding or excluded strand interactions. This needs to be in the Discussion.

Reviewer #2 (Remarks to the Author):

The manuscript by Kose et al. provides important fundamental insight about the architecture of the DNA-unwinding component of the eukaryotic replication fork. The center of this "unwinding fork" consists of a ring of 6 MCM proteins that are highly similar in sequence and structure. It is widely accepted that the MCM ring uses a strand exclusion mechanism to separate the constituent strands of the DNA double-helix so that each individual strand can be used as a template in the synthesis of new DNA. In this strand exclusion mechanism, the MCM ring encircles the leading strand DNA template and excludes the lagging strand DNA template. This topological arrangement is then able to propagate strand separation when the ring moves in the 3' to 5' direction along the encircled leading strand template DNA. Although the movement of the MCM ring on the encircled DNA strand is mechanistically well-understood, how this movement specifically causes the two strands to separate is not. The current manuscript advances the mechanistic picture of DNA unwinding.

The 3' to 5' movement of the MCM ring along the encircled strand (termed "DNA translocation") has previously been illustrated with a crystal structure that shows how the MCM ring interacts with the encircled strand and also how the MCM DNA-binding elements move in response to ATP hydrolysis at the MCM ATPase sites (PDB 6MII). In contrast, the specific arrangement of a fork DNA structure during unwinding has not been fully defined because the 5'-arm of the fork is disordered in the EM structure of the ring bound to fork DNA (PDB 5U8S). This ambiguity allows the possibility that the ring has encircled both DNA strands in the structure-- which would not be the topological form used to unwind DNA. Notably, an arrangement where the MCM ring encircles two opposite polarity single-strands is well-precedented-- the archaeal MCM of *Sulfolobus solfataricus* binds bubble substrates with higher affinity than single-stranded DNA and also double-stranded DNA (J Biol Chem. 2004 Nov 19;279(47):49222-8). These affinities imply that the ring has a preference to encircle hybrid single/double-stranded substrates with two single-strands opposite in polarity—which also exists in fork DNA structures to potentially confuse biochemical analysis.

The present manuscript addresses this scenario with conditions expected to prevent the ring from encircling the 5'-arm of the fork, such as the addition of a strand complementary to the 5'-arm ("dsLag"), or addition of an adduct to the 5'-arm. All of these experiments provide a consistent picture that steric bulk on the 5'-arm of the fork accelerates unwinding. In interpreting these results, the manuscript focusses on transient species where the 5' arm partially enters the channel as the underlying reason that the helicase is slow with naked 5'-arms. However, it is also possible that when a naked 5'-arm is used, a fraction of the MCM rings fully encircles both DNA strands at the outset (see above) and thus are not in the "strand excluded" topology needed to unwind DNA,

and that steric bulk on the 5'-arm reduces this fraction. This scenario is conceptually very similar to that described in the manuscript, but it would have fundamental implications for the biochemical study of MCM helicases. The presence of this scenario would not detract from the significance of the manuscript.

Additional comments:

The authors ascribe the differences between CMG and T-antigen to different sizes of the ring channel. In the manuscript's present form, this is subjective and may not be true depending on the channel criteria. A plot is enclosed to illustrate the minimum radius along the length of the channel for the two families of ring helicases according to one set of criteria (see plot legend). In general, the channel width is highly dependent on the nucleotide state (4R7Y vs. 6MII and also 1SVL vs. 1SVM). Of the rings plotted, the narrowest constriction occurs at the helix-2-insert hairpins of MCM in 6MII. A different set of criteria may more faithfully illustrate the intended meaning.

I think the action of CompLag in Fig. 6b could also be interpreted to occur in the wake of the helicase as described for Fig. 2c and d.

For this plot, each hexamer was aligned to place the axis that permutes the OB-folds (MCM) or the oligomerization domains (E1 & T-antigen) on the Z-axis. The models were all converted to poly-alanine to discount side-chains that could be flexible, and the distance of each atom to the channel axis was calculated as $\sqrt{X^2 + Y^2}$. The minimum distance in 2-angstrom wide bins along the Z-coordinate was plotted against the Z-coordinate to give a minimum channel radius as a function of the channel length.

We thank both reviewers for stating that “the experimental quality is excellent” and that “it provides important fundamental insight about the architecture of the DNA-unwinding component of the eukaryotic replication fork giving constructive feedback”. We appreciate reviewers’ comments, which helped us improve our manuscript significantly. Below we provide point-by-point response to concerns raised by the reviewers.

Reviewer #1 (Remarks to the Author):

Review of “Duplex DNA engagement and a replisome factor oppositely regulate the speed of a replicative DNA helicase” by Kose et al. The authors have investigated the interaction of the eukaryotic CMG helicase with both strands of DNA as mediated by the single stranded binding protein, RPA, using both bulk and single molecule assays. They provide a plausible explanation of the observed stalling when naked ssDNA is the excluded strand that suggests duplex DNA enters the central pore entering a nonproductive state. RPA binding (or hairpin DNA) to the excluded strand prevents this pause and stimulates unwinding. The experimental quality is excellent, however there are few additional experiments that are needed to better support their conclusions and eliminate alternative hypotheses.

1) Figure 1- An important control would be to monitor CMG unwinding in the absence of RPA using the same experimental setup to directly compare the rates of unwinding.

Performing the single-molecule TIRF-based assay without RPA to monitor CMG unwinding duplex DNA is technically challenging. Our results in Figure 2 and with magnetic-tweezers (Burnham et al. 2019 Nat. Comm.) indicate that CMG unwinds duplex DNA at an average rate of approximately 0.2 bps^{-1} as opposed to 4.5 bps^{-1} in the presence of RPA (Figure 1e). Therefore, in order to visualize CMG unwind 5kb dsDNA (i.e. half the length of 10kb DNA used in Figure 1) we would have to perform image acquisition for approximately 7 hours. Relatively poor spatial resolution due to stage drift and difficulties visualizing fluorescently-labelled CMG due to photobleaching make it impossible to observe CMG movement for such a long duration and determine helicase speed in RPA’s absence using the TIRF-based assay.

2) Line 157- “extensive” unwinding is stated with RPA, but Figure 2a, lane 4, shows less than 50% unwound. A better way to state this would be ‘increased’ unwinding compared to lane 3? (also number of the lanes in Figure 2a, c, and d would help)

We thank the reviewer to point this out. We changed the phrase “extensive unwinding” to “significant unwinding”. We now also numbered lanes below gel images in Figure 2.

3) A control is needed to show that RPA cannot displace the stable hairpin and bind resulting ssDNA that is routinely used on the lagging strand.

It is possible that RPA may bind the hairpin on the lagging-strand tail of the fork substrate used in Figure 2c. The main reason for designing the lagging-strand tail to make a hairpin on this substrate was to prevent helicase stalling at the fork junction. Since RPA binding to the lagging-strand tail also prevents CMG from pausing, whether the lagging-strand tail is a hairpin or RPA-bound ssDNA should not change the outcome of our results.

4) In Figure 2d, could competitor DNA complementary to the translocating strand increase the unwinding rate as well. Or is it specifically dependent on RPA.

We would like to clarify the use of RPA and competitor DNA in Figures 2 c and d. In Figure 2d competitor oligonucleotide, which is complementary to the translocating strand, is added together with ATP only to block reannealing of Cy5-modified strand to the translocating strand after CMG

runs off the 5' end. In Figure 2d, no RPA is added to the reaction. In Figure 2c, RPA is added to the reaction with ATP. Since RPA already impedes reannealing of the displaced Cy5-labelled strand, addition of competitor oligonucleotide is not needed.

Importantly, the competitor oligonucleotide in Figure 2d is complementary only to the 28-nt stretch on the 5' end of the translocating strand. Because the majority of CMG translocation occurs through the upstream 252-bp stretch of duplex DNA, any increase in the rate of unwinding within 28-bp DNA would minimally affect the overall unwinding rate.

5) That said, could other SSBs (gp32 or SSB) increase the unwinding rate similar to RPA or is it species dependent.

Overall, our data suggest that RPA increases the rate of CMG-driven DNA unwinding by blocking the helicase to engage with duplex DNA at the fork junction. Given that even a streptavidin bound to biotin on the excluded strand is sufficient to prevent duplex engagement by CMG (Figure 5c), other SSBs are expected to increase the rate of DNA unwinding by CMG. In fact, using single-molecule magnetic tweezers, we previously showed that *E.coli* SSB speeds up the CMG helicase about 10-fold (can be seen in the online peer review file of Burnham et al. 2019 Nat. Commun.) consistent with our results with RPA (Figures 1 and 2). Since other SSBs are not relevant to the eukaryotic replisome, we chose not to include any data with CMG in the presence of *E.coli* SSB.

6) Moreover, in later Figures 3-5, you show that RPA increases the initiation of unwinding which can lead to an increased rate of unwinding (Fig 5). Therefore, experiments in Figure 2 should be repeated with RPA bound prior to initiation with CMG to compare directly with Fig 2c.

In Figures 3 and 4, we used fork DNA substrates containing single stranded 5' tails to show that duplexing the 5' lagging-strand tail of short fork DNA substrates increases the efficiency of unwinding most likely by restricting CMG to pause at the fork junction. Similarly, Supplementary Figure 3a demonstrates that hairpin-like secondary structures on the lagging-strand tail prevents helicase pausing leading to higher unwinding efficiency. Since all DNA substrates used in Figure 2 make hairpin structures on the lagging-strand tail that avoids CMG pausing at the fork junction, pre-binding of RPA is not needed to stimulate initiation. Consistent with this, we found that CMG-dependent DNA unwinding kinetics were essentially the same when RPA was added prior to ATP to those when RPA was added together with ATP (see Figure below).

The experiment in Figure 2c was performed with the exception of adding RPA 3 minutes prior to ATP. The bands were quantified and plotted as a function of time. Addition of RPA prior to ATP (red squares) did not change the overall kinetics of CMG-mediated DNA unwinding. The quantification shown in Figure 2e is also shown for comparison (blue circles).

7) Figure 4a, it seems possible that 50% of the CMG molecules also bind the 5' arm, thus rendering them inactive for unwinding. That is the reason you see a 2-fold decrease in unwound product compared to dsLag. This could also explain the results in Figure 5a where there is a 3-fold difference in fluorescence intensity.

It is very unlikely that the observed difference in unwinding efficiencies between Fork^{ssLag} and Fork^{dsLag} in Figure 4 is due to more efficient CMG binding to the 3' tail of Fork^{dsLag}. In these single-molecule experiments, fork DNA was immobilised to the surface and excess DNA was removed

before CMG was introduced. Since effective DNA concentration is extremely low (practically negligible), excess CMG present in solution would not be competed out by excess DNA. Therefore, unless binding of CMG to the 5' tail prevents binding of another CMG to the 3' tail, the observed difference in unwinding could not be due to CMG binding. We note that it has been previously shown that two CMG molecules can bind both 3' and 5' tails of fork DNA simultaneously (Petojevic et al. PNAS 2015). Importantly, we showed that when CMG was pre-bound to Fork^{ssLag}, addition of an oligonucleotide complementary to the 5' tail together with ATP was sufficient to increase DNA unwinding efficiency in bulk assays (Supplementary Figure 3e). This data demonstrates that the increase in DNA unwinding efficiency when the 5' tail is duplexed is not due to more efficient CMG binding. The equivalent single-molecule assay would be to bind CMG to the surface-immobilised Fork^{ssLag}, wash free CMG, and introduce an oligonucleotide complementary to the lagging-strand arm before ATP addition. We tried this experiment, however removing free CMG with a buffer wash of the flow channel before ATP addition prevented efficient DNA unwinding regardless of the DNA substrate (either Fork^{ssLag} or Fork^{dsLag}) used for reasons that are not clear at this time.

8) Figure 4c, it is not clear where exactly the biotin is incorporated on the ssLag. Is it in a position to prevent your model of ds engagement for stalling? It would be useful to vary the position of the biotin (on the Lag strand, at the duplex junction, and then several bases into the duplex) to test this model. Need to expand and test other biotin locations in combination with Figure 7.

We believe the reviewer means Figure 5c. We thank the reviewer for the remark about biotin position. It was an oversight on our part not to clarify the position of biotin on the lagging-strand arm of this fork DNA substrate. On the original substrate we used, the biotin moiety was located on the 5th nucleotide from the ssDNA-dsDNA junction. Therefore, it is unlikely that streptavidin at this position stimulates unwinding by preventing the outer surface of CMG from interacting with the lagging-strand arm. To rule out this possibility further, we designed a fork DNA substrate, which contained a biotin on the lagging-strand arm adjacent to the ssDNA-dsDNA junction. Addition of streptavidin similarly increased the efficiency of CMG-mediated DNA unwinding. We now replaced the data with this new substrate and clarified the position of biotin in the text.

9) Figure 5a, Why doesn't the addition of the CompLag DNA restore unwinding as in 5b?

10) And in this experiment 5a, is the CompLag only complementary to the 5' arm or parts of the duplex region as well? This is not clear based on your discussion or in the Figure. At the bottom of the blue scheme in 5a, it appears only complementary to the 5' arm. This appears different than in 5b, where DNALag has complementary to the duplex excluded strand.

11) Points 9 and 10 above are important to understand prior to the interpretation of the results as it may explain the differences in the red traces for both. It is frustrating that the authors did not better explain this important facet of the experiment.

We will address points 9 through 11 together as they are part of the same concern. First, we believe the reviewer is referring to Figures 6 a and b (not Figures 5 a and b). We apologize for not conveying the experimental design clearly. The main difference between the experiments shown in Figures 6 a and b is that, while Comp^{Lag} in Figure 6a is complementary only to the single-stranded 5' tail of fork substrate, Comp^{Lag} used in 6b is complementary to the lagging-strand only within the parental duplex region. To make this discrepancy more obvious, we now renamed complementary oligonucleotides as Comp^{LagTail} for Figure 6a and or Comp^{LagParent} for Figure 6b.

Our data suggests that when CMG pauses upon engaging with the parental duplex, DNA reannealing is required to push CMG backwards to free the helicase from this paused state. If CMG pauses at the fork junction on the substrate used in Figure 6a, it gets permanently trapped since the two non-complementary arms of this fork can not anneal. Thus, adding Comp^{LagArm} after ATP does not rescue unwinding. The substrate used in Figure 6b has two major differences to that used in 6a. The first is

that the lagging-strand arm makes a hairpin-like structure preventing CMG from pausing at the fork junction. The second is that it has a longer duplex region (60 bp as opposed to 28 bp) such that CMG may enter the paused state at sites within the parental duplex. If CMG enters such a paused state within the 60-bp duplex, the two complementary strands within this region should be able to reanneal pushing the helicase backwards and freeing it from the paused state. According to this model, adding $\text{Comp}^{\text{LagParent}}$ after ATP should be able to rescue unwinding (in contrast to $\text{Comp}^{\text{LagArm}}$ in 6a). Indeed, adding $\text{Comp}^{\text{LagParent}}$ after ATP leads DNA unwinding levels to quickly recover (Figure 6b, $\text{ATP} \rightarrow \text{Comp}^{\text{LagParent}}$). We now revised the text in the manuscript to further clarify our experiments.

12) Figure 7 may be useful in exploiting SA to distinguish between duplex sliding and excluded strand engagement, however, as is, it is not complete. There is no difference in the unwinding rates +/-SA, but this substrate (ssDNA 5' tail) is the one that should show slow unwinding. This experiment needs 1) repeat with either duplex 5'-tail or hairpin, 2) addition of complementary DNALag and 3) addition of and effect of RPA as well as some combination of all the 3 points.

The purpose of the experiment shown in Figure 7 is not to test whether a protein on the lagging-strand template prevents duplex engagement. Here, we aim to demonstrate that a lagging-strand protein barrier on the duplex region does not hamper CMG translocation. Earlier work by the O'Donnell group suggested that a protein bound to the lagging-strand template inhibits DNA unwinding by CMG (Langston et al. 2017 eLife). This led to the proposal that CMG may encircle a short stretch of duplex DNA at the fork junction while actively unwinding DNA. We later demonstrated that CMG can efficiently bypass lagging-strand protein roadblocks (Kose et al. 2019 Cell Rep.). A recent paper from the O'Donnell group suggested that the long linker between DNA and the protein barrier present in Kose et al. 2019 study may account for the efficient bypass of lagging-strand barriers by CMG (Yuan et al. 2020 Nat. Commun.). In Figure 7, we show that CMG is able to efficiently bypass a protein barrier which was directly crosslinked to a single base on the lagging-strand template without a linker. These results indicate that actively translocating CMG does not encircle a stretch of dsDNA in its interior channel.

We thank the reviewer for pointing out the lagging-strand arm on this fork substrate. The cartoon drawing on Figure 7b was incorrect. In fact, the 5' tail of this DNA substrate contained d(GGCA)₁₀ sequence that folds into hairpin-like secondary structures, which prevent CMG to pause at the fork junction. We now corrected cartoon in the figure and clarified the sequence in the figure legend.

13) I am not sure any of your experiments can absolutely eliminate the possibility that the ssLag can interact with the exterior of the CMG and slow unwinding. If the SA experiments (Fig 5c) are designed correctly, then this may be good support, however, it is not clear that the biotin is in the correct position to absolutely support your conclusion.

As described in our response to point 8, the fork DNA used in Figure 5c contained a biotin that was located 5 nucleotides away from the ss-dsDNA junction. Therefore, it is unlikely that streptavidin bound to the biotin at this position stimulates unwinding by preventing the outer surface of CMG to interact with the lagging-strand arm. To rule out this possibility further, we designed a fork DNA substrate which contained a biotin on the lagging-strand arm one nucleotide from the fork junction. Addition of streptavidin similarly increased the efficiency of CMG-mediated DNA unwinding on this substrate. We now replaced the data with this new substrate (Figure 5c). Together, we believe that our results strongly support a model in which a protein on the lagging-strand arm stimulates CMG helicase activity by obstructing CMG from interacting with the parental duplex at the fork junction.

14) There are no direct measurements or monitoring of the duplex DNA region during a purported stalled state. All experiments are indirect and could be explained by an excluded strand interaction.

We refer to our answer in point 13. We now show that streptavidin bound to the lagging-strand arm adjacent to the fork junction is able to increase the extent of DNA unwinding by CMG (Figure 5c). Streptavidin at this position should not inhibit a potential interaction between CMG and the excluded

strand. Therefore, our data is more consistent with a model where CMG interaction with the fork junction leads the helicase to pause.

15) Figure 8: this cartoon is used to explain your model, but nowhere did you test the RPA exhaustion model. You would need to test unwinding under different RPA concentrations or pulse chase experiments. Otherwise, Figure 8 should stick to the idea of duplex sliding/stalling, and/or whether you can fully eliminate excluded strand interaction.

We removed the cartoon from Figure 8 in which RPA exhaustion leads to helicase pausing. We now show two different scenarios:

Figure 8a. In the absence of RPA, CMG unwinds DNA slowly due to frequent pausing

Figure 8b. RPA prevent duplex engagement by CMG leading to faster DNA unwinding

Minor Points

1) Figure supplement 1c, there appears to be a second Cy3 red star on the DNA in the bottom panel of the cartoon which shouldn't be there.

We thank the reviewer to indicate this mistake. We removed the inaccurately-placed red star representing a second Cy3 from Supplementary Figure 1c.

2) MCM10 and CTF4 are mentioned exactly once in the introduction. Based on 2 recent papers from the O'Donnell group they may also have a role in preventing or regulating duplex sliding or excluded strand interactions. This needs to be in the Discussion.

In the original discussion, we stated that “Furthermore, like RPA, other replication factors such as polymerase alpha-primase, Mcm10, and AND-1 (Ctf4), may also promote DNA unwinding by keeping the helicase from interacting with the parental duplex”. We now also added the following statement to the discussion: “Mcm10, which is proposed to open a gate in the CMG complex, may also facilitate displacement of duplex DNA from the helicase central channel when needed (Wasserman et al. 2019 Cell)”. Here, the article from Shixin Liu and Michael O'Donnell groups is cited. The recent work on Ctf4 from Huilin Li and Michael O'Donnell laboratories showed that Ctf4 can dimerize CMG complexes (Yuan et al. 2019 eLife). We do not think this study is relevant to our work.

Reviewer #2 (Remarks to the Author):

The manuscript by Kose et al. provides important fundamental insight about the architecture of the DNA-unwinding component of the eukaryotic replication fork. The center of this “unwinding fork” consists of a ring of 6 MCM proteins that are highly similar in sequence and structure. It is widely accepted that the MCM ring uses a strand exclusion mechanism to separate the constituent strands of the DNA double-helix so that each individual strand can be used as a template in the synthesis of new DNA. In this strand exclusion mechanism, the MCM ring encircles the leading strand DNA template and excludes the lagging strand DNA template. This topological arrangement is then able to propagate strand separation when the ring moves in the 3' to 5' direction along the encircled leading strand template DNA. Although the movement of the MCM ring on the encircled DNA strand is mechanistically well-understood, how this movement specifically causes the two strands to separate is not. The current manuscript advances the mechanistic picture of DNA unwinding.

The 3' to 5' movement of the MCM ring along the encircled strand (termed “DNA translocation”) has previously been illustrated with a crystal structure that shows how the MCM ring interacts with the encircled strand and also how the MCM DNA-binding elements move in response to ATP hydrolysis at the MCM ATPase sites (PDB 6MII). In contrast, the specific arrangement of a fork DNA structure during unwinding has not been fully defined because the 5'-arm of the fork is disordered in the EM

structure of the ring bound to fork DNA (PDB 5U8S). This ambiguity allows the possibility that the ring has encircled both DNA strands in the structure-- which would not be the topological form used to unwind DNA. Notably, an arrangement where the MCM ring encircles two opposite polarity single-strands is well-precedented-- the archaeal MCM of *Sulfolobus solfataricus* binds bubble substrates with higher affinity than single-stranded DNA and also double-stranded DNA (J Biol Chem. 2004 Nov 19;279(47):49222-8). These affinities imply that the ring has a preference to encircle hybrid single/double-stranded substrates with two single-strands opposite in polarity—which also exists in fork DNA structures to potentially confuse biochemical analysis.

The present manuscript addresses this scenario with conditions expected to prevent the ring from encircling the 5'-arm of the fork, such as the addition of a strand complementary to the 5'-arm ("dsLag"), or addition of an adduct to the 5'-arm. All of these experiments provide a consistent picture that steric bulk on the 5'-arm of the fork accelerates unwinding. In interpreting these results, the manuscript focusses on transient species where the 5' arm partially enters the channel as the underlying reason that the helicase is slow with naked 5'-arms. However, it is also possible that when a naked 5'-arm is used, a fraction of the MCM rings fully encircles both DNA strands at the outset (see above) and thus are not in the "strand excluded" topology needed to unwind DNA, and that steric bulk on the 5'-arm reduces this fraction. This scenario is conceptually very similar to that described in the manuscript, but it would have fundamental implications for the biochemical study of MCM helicases. The presence of this scenario would not detract from the significance of the manuscript.

We appreciate the reviewer's remark about the possibility that inefficient unwinding of Fork^{ssLag} could be due to a fraction of CMG complexes fully encircling both strands. Indeed, this scenario could explain the data presented in Figure 3. However, if this was the case adding the oligonucleotide complementary to the lagging-strand tail post-CMG binding is not expected to stimulate unwinding. The data we presented in Supplementary Figure 3e strongly argues against a model where enhanced unwinding of Fork^{dsLag} is due to more efficient/proper binding of CMG to the 3' tail of the fork.

Additional comments:

The authors ascribe the differences between CMG and T-antigen to different sizes of the ring channel. In the manuscript's present form, this is subjective and may not be true depending on the channel criteria. A plot is enclosed to illustrate the minimum radius along the length of the channel for the two families of ring helicases according to one set of criteria (see plot legend). In general, the channel width is highly dependent on the nucleotide state (4R7Y vs. 6MII and also 1SVL vs. 1SVM). Of the rings plotted, the narrowest constriction occurs at the helix-2-insert hairpins of MCM in 6MII. A different set of criteria may more faithfully illustrate the intended meaning.

We thank the reviewer for pointing this out. We now changed the statement "*This result suggests that regulation of DNA unwinding by replicative helicases through duplex DNA interaction at the fork depends on the size of their central channel*" in lines 242-244 to "*This result suggests that regulation of DNA unwinding by replicative helicases through duplex DNA interaction at the fork correlate with their ability to translocate on dsDNA.*"

Accordingly, we revised the following sentence in the Discussion (lines 484-486) from "*In contrast, large T antigen, which has a relatively small central channel, unwinds dsDNA only 2-fold slower than it translocates on ssDNA.*" to "*In contrast, large T antigen, which does not slide onto dsDNA upon meeting a flush ss-dsDNA junction, unwinds dsDNA only 2-fold slower than it translocates on ssDNA.*"

Finally, we changed the sentence in lines 488-489 from "*...the likelihood of helicase slowing down at the fork correlates with their central pore size*" to "*...the likelihood of helicase slowing down at the fork correlates with their capacity to translocate along dsDNA.*"

I think the action of CompLag in Fig. 6b could also be interpreted to occur in the wake of the helicase as described for Fig. 2c and d.

We thank the reviewer for bringing this possibility to our attention. To test whether stimulation of DNA unwinding by Comp^{LagParent} in Figure 6b is solely due to prevention of fork DNA hybridization behind the helicase, we tested how an oligonucleotide complementary to the leading-strand template within the parental duplex (Comp^{LeadParent}) affects DNA unwinding efficiency. We found that while Comp^{LeadParent} also increased the efficiency of CMG-driven DNA unwinding, stimulation was significantly lower compared to addition of Comp^{LagParent}. This data is now presented in Supplementary Figure 6. Therefore, our data suggests that increased DNA unwinding efficiency by Comp^{LagParent} can not be solely attributed to inhibition of DNA reannealing behind the helicase.

REVIEWERS' COMMENTS:

Reviewer #1 (Remarks to the Author):

Second review of "Duplex DNA engagement and a replisome factor oppositely regulate the DNA unwinding rate of CMG helicase"

After reading the responses to the comments from the original review, I have a better understanding of the main conclusions of the manuscript that 1) CMG routinely pauses at a fork junction by engaging with the duplex and 2) replication factors, like RPA can stimulate a reversal reannealing mode that can reactivate CMG for unwinding. It is now more focused on the central results and limits the discussion to relevant findings.

1) That said, I am concerned about the "replication factor" terminology in the title. It appears vague. I realize that the authors are not trying to limit this reengagement activity to only RPA, but maybe they should in the title as RPA is what they have primarily been using to show the increased unwinding activity.

2) Reresponding to Reviewer 1, original comment 3 and their response. My only further concern with this is that the authors should differentiate RPA as a CMG loader/initiator, versus RPA as a CMG rate enhancer. Maybe that question can be addressed with a different assay (see point 5 below), but they have never fully resolved this question. If they compared the effect of a complementary (trap) DNA strand to RPA, this may help to clarify its role.

In my opinion, the appropriate control in Figure 2a would be to have RPA alone in the presence of ATP and no CMG. That way you can calculate the CMG dependent unwinding and determine whether there is any background unwinding by RPA. (They have an RPA alone control in Supplementary Figure 2a on a slightly different substrate. RPA alone (orange trace) shows some background unwinding).

3) Reresponding to Reviewer 1, original comment 4 and their response. (this is also related to point 2 above). Figure 2d, Can a competitor strand complementary to the translocating strand (whole length 252bp+28 or 252bp duplex) prevent strand re-annealing in Figure 2d?. If yes, the unwinding rates should increase, may be to a similar rate as observed in RPA. Having this new competitor strand determines whether the increase in unwinding rate seen in figure 2C is unique to RPA or can something else (like the competitor strand) impede strand reannealing and increase the unwinding rate.

4) Reresponding to Reviewer 1, original comment 5 and their response. Some aspect of their original response would be worthwhile to state clearly in the manuscript text.

Overall, our data suggest that RPA increases the rate of CMG-driven DNA unwinding by blocking the helicase to engage with duplex DNA at the fork junction. Given that even a streptavidin bound to biotin on the excluded strand is sufficient to prevent duplex engagement by CMG (Figure 5c), other SSBs are expected to increase the rate of DNA unwinding by CMG. In fact, using single-molecule magnetic tweezers, we previously showed that E.coli SSB speeds up the CMG helicase about 10-fold (can be seen in the online peer review file of Burnham et al. 2019 Nat. Commun.) consistent with our results with RPA (Figures 1 and 2). Since other SSBs are not relevant to the eukaryotic replisome, we chose not to include any data with CMG in the presence of E.coli SSB

5) Reresponding to Reviewer 1, original comment 6 and their response.

The conditions in Figure 5 that I was interested in was the fact that they pre bound RPA, then washed away free RPA with competitor, so that the RPA only assists CMG in initiation, and not in elongation. This would help to differentiate loading assist vs rate assist. I agree with the authors that prebinding RPA does not seem to matter with the presence of free RPA in solution. However the question was not about whether prebinding is important, it was about whether or not RPA bound to the substrate only at the beginning of the experiment was not allowed to assist with later fork junctions would cause a measurable increase in unwinding.

Reviewer #2 (Remarks to the Author):

The authors have considered and addressed my points of concern with the original submission. I have 2 responses that the authors may want to consider in finalizing the manuscript.

1. To respond to the possibility that different rates of unwinding might be due to different "topological populations" where a larger fraction is in a "strand-excluded" topology with CompLagTail than without, the authors argue that CompLagTail is added after the CMG-fork DNA species has been generated in the ATP-gS step (Fig. S-3e). This argument includes a premise that the "ring:DNA topology population" cannot change significantly following the ATP-gS step (for example that addition of CompLagTail does not alter the distribution). I am not certain this is known, but it may be. This premise and its justification should be provided in a quick sentence or two.

2. The removal of the discussion of ring channel diameters has addressed my other point of concern. I think one of the new sentences is potentially confusing:

"These results support a model whereby inhibition of DNA unwinding due to parental duplex engagement is a shared characteristic among replicative helicases "

This sentence directly follows a sentence that states that SV40 T-antigen does not engage dsDNA-- suggesting that T-antigen actually does not share this characteristic. I think the discussion is intended to say that SV40 T-antigen is not subject to this characteristic because it lacks dsDNA translocation activity. This area could be worded more precisely to say that it "may be a shared characteristic among the replicative helicases that are able to translocate dsDNA."

REVIEWERS' COMMENTS:

Reviewer #1 (Remarks to the Author):

Second review of “Duplex DNA engagement and a replisome factor oppositely regulate the DNA unwinding rate of CMG helicase”

After reading the responses to the comments from the original review, I have a better understanding of the main conclusions of the manuscript that 1) CMG routinely pauses at a fork junction by engaging with the duplex and 2) replication factors, like RPA can stimulate a reversal reannealing mode that can reactivate CMG for unwinding. It is now more focused on the central results and limits the discussion to relevant findings.

1) That said, I am concerned about the “replication factor” terminology in the title. It appears vague. I realize that the authors are not trying to limit this reengagement activity to only RPA, but maybe they should in the title as RPA is what they have primarily been using to show the increased unwinding activity.

We now replaced the “replication factor” with “RPA” in the title.

2) Reresponding to Reviewer 1, original comment 3 and their response. My only further concern with this is that the authors should differentiate RPA as a CMG loader/initiator, versus RPA as a CMG rate enhancer. Maybe that question can be addressed with a different assay (see point 5 below), but they have never fully resolved this question. If they compared the effect of a complementary (trap) DNA strand to RPA, this may help to clarify its role. In my opinion, the appropriate control in Figure 2a would be to have RPA alone in the presence of ATP and no CMG. That way you can calculate the CMG dependent unwinding and determine whether there is any background unwinding by RPA. (They have an RPA alone control in Supplementary Figure 2a on a slightly different substrate. RPA alone (orange trace) shows some background unwinding).

Original comment 3: 3) A control is needed to show that RPA cannot displace the stable hairpin and bind resulting ssDNA that is routinely used on the lagging strand.

The reviewer suggests that “the appropriate control in Figure 2a would be to have RPA alone in the presence of ATP and no CMG”. We included a control in Figure 2a with RPA and no CMG that shows no detectable unwinding of 236-bp of duplex DNA. RPA is not known to melt DNA in an ATP-dependent manner, therefore we did not include an additional control with ATP and RPA in the absence of CMG for this substrate. As the reviewer pointed out, there is some background unwinding by RPA alone for the substrate used in Figure 2c (as measured in Supplementary Figure 2a). We believe this is due to the relatively short size of duplex DNA (28 bp) to be unwound. For this reason, the data in Figure 2e, which is the quantification of Figure 2c, was corrected for CMG-independent unwinding by RPA using

data from Supplementary Figure 2a (orange trace).

3) Reresponding to Reviewer 1, original comment 4 and their response. (this is also related to point 2 above). Figure 2d, Can a competitor strand complementary to the translocating strand (whole length 252bp+28 or 252bp duplex) prevent strand re-annealing in Figure 2d?. If yes, the unwinding rates should increase, may be to a similar rate as observed in RPA. Having this new competitor strand determines whether the increase in unwinding rate seen in figure 2C is unique to RPA or can something else (like the competitor strand) impede strand reannealing and increase the unwinding rate.

Original comment 4: In Figure 2d, could competitor DNA complementary to the translocating strand increase the unwinding rate as well. Or is it specifically dependent on RPA.

This is an interesting point. Unfortunately, such a long oligonucleotide (252 nt) would form secondary structures and will most likely not efficiently anneal to the unwound fork template. Therefore, we have not tried adding a complementary oligonucleotide to the 252-bp duplex region.

4) Reresponding to Reviewer 1, original comment 5 and their response. Some aspect of their original response would be worthwhile to state clearly in the manuscript text. Overall, our data suggest that RPA increases the rate of CMG-driven DNA unwinding by blocking the helicase to engage with duplex DNA at the fork junction. Given that even a streptavidin bound to biotin on the excluded strand is sufficient to prevent duplex engagement by CMG (Figure 5c), other SSBs are expected to increase the rate of DNA unwinding by CMG. In fact, using single-molecule magnetic tweezers, we previously showed that E.coli SSB speeds up the CMG helicase about 10-fold (can be seen in the online peer review file of Burnham et al. 2019 Nat. Commun.) consistent with our results with RPA (Figures 1 and 2). Since other SSBs are not relevant to the eukaryotic replisome, we chose not to include any data with CMG in the presence of E.coli SSB.

We now included the following sentence in the discussion:

*“We note that *E.coli* single-stranded binding (SSB) protein also speeds up CMG-driven duplex unwinding (Burnham et al.)”*

5) Reresponding to Reviewer 1, original comment 6 and their response. The conditions in Figure 5 that I was interested in was the fact that they pre bound RPA, then washed away free RPA with competitor, so that the RPA only assists CMG in initiation, and not in elongation. This would help to differentiate loading assist vs rate assist. I agree with the authors that prebinding RPA does not seem to matter with the presence of free RPA in solution. However the question was not about whether prebinding is important, it was about whether or not RPA bound to the substrate only at the beginning of the experiment was not allowed to assist with later fork junctions would cause a measurable increase in unwinding.

In the original comment 6, the reviewer stated: "Experiments in Figure 2 should be repeated with RPA bound prior to initiation with CMG to compare directly with Fig 2c". We have performed this experiment and provided the results in our first rebuttal.

It is unlikely that RPA pre-bound to the 5' tail on the fork substrate used in Figure 5a increases the speed of CMG translocation through the duplex region in addition to stimulation of initiation. Because the duplex region on this fork DNA substrate is relatively short (28 bp), measuring the rate of CMG translocation reliably is not possible.

Reviewer #2 (Remarks to the Author):

The authors have considered and addressed my points of concern with the original submission. I have 2 responses that the authors may want to consider in finalizing the manuscript.

1. To respond to the possibility that different rates of unwinding might be due to different "topological populations" where a larger fraction is in a "strand-excluded" topology with CompLagTail than without, the authors argue that CompLagTail is added after the CMG-fork DNA species has been generated in the ATP-gS step (Fig. S-3e). This argument includes a premise that the "ring:DNA topology population" cannot change significantly following the ATP-gS step (for example that addition of CompLagTail does not alter the distribution). I am not certain this is known, but it may be. This premise and its justification should be provided in a quick sentence or two.

We now added the following statement (lines 230-233): "We cannot exclude the possibility that CMG may encircle both the 3'- and 5'-ssDNA tails in its central channel when bound to the fork substrate with ATPgS and that adding Comp^{LagTail} may configure CMG to exclude the 5' tail from the helicase pore".

2. The removal of the discussion of ring channel diameters has addressed my other point of concern. I think one of the new sentences is potentially confusing:

"These results support a model whereby inhibition of DNA unwinding due to parental duplex engagement is a shared characteristic among replicative helicases "

This sentence directly follows a sentence that states that SV40 T-antigen does not engage dsDNA-- suggesting that T-antigen actually does not share this characteristic. I think the discussion is intended to say that SV40 T-antigen is not subject to this characteristic because it lacks dsDNA translocation activity. This area could be worded more precisely to say that it "may be a shared characteristic among the replicative helicases that are able to translocate dsDNA."

While large T antigen does not translocate along dsDNA, it may still engage with duplex DNA at the fork junction to some extent. Accordingly, while the effect of duplexing the lagging-strand arm on T antigen-mediated unwinding is not as great as that on CMG, there is still some stimulation in helicase activity (Figure 3e, 10-fold fluorescence increase for CMG and

2-fold increase for large T antigen). We now changed our statement regarding T antigen in the discussion from

“In contrast, large T antigen, which does not slide onto dsDNA upon meeting a flush ss-dsDNA junction, unwinds dsDNA only 2-fold slower than it translocates on ssDNA. These results support a model whereby inhibition of DNA unwinding due to parental duplex engagement is a shared characteristic among replicative helicases and suggests that the likelihood of helicase slowing down at the fork correlates with their capacity to translocate along dsDNA.”

to

“Large T antigen, another ring-shaped replicative helicase, unwinds dsDNA only 2-fold slower than it translocates on ssDNA. While large T antigen does not slide onto dsDNA upon meeting a flush ss-dsDNA junction, it may still interact with the parental duplex at the fork junction to some extent. This idea is in line with our observation that duplexing the lagging-strand arm of fork DNA stimulates T antigen helicase activity, albeit to a lesser degree than CMG (Figure 3). Together, our results support a model whereby inhibition of DNA unwinding due to parental duplex engagement is a shared characteristic among replicative helicases and suggests that the likelihood of helicase slowing down at the fork correlates with their capacity to translocate along dsDNA.”.